# AC-GC: Lossy Activation Compression with Guaranteed Convergence

**R. David Evans**
Dept. of Electrical and Computer Engineering
University of British Columbia
Vancouver, BC V6T 1Z4
rdevans@ece.ubc.ca

**Tor M. Aamodt**
Dept. of Electrical and Computer Engineering
University of British Columbia
Vancouver, BC V6T 1Z4
aamodt@ece.ubc.ca

## Abstract

Parallel hardware devices (e.g., graphics processor units) have limited high-bandwidth memory capacity. This negatively impacts the training of deep neural networks (DNNs) by increasing runtime and/or decreasing accuracy when reducing model and/or batch size to fit this capacity. Lossy compression is a promising approach to tackling memory capacity constraints, but prior approaches rely on hyperparameter search to achieve a suitable trade-off between convergence and compression, negating runtime benefits. In this paper we build upon recent developments on Stochastic Gradient Descent convergence to prove an upper bound on the expected loss increase when training with compressed activation storage. We then express activation compression error in terms of this bound, allowing the compression rate to adapt to training conditions automatically. The advantage of our approach, called AC-GC, over existing lossy compression frameworks is that, given a preset allowable increase in loss, significant compression without significant increase in error can be achieved with a single training run. When combined with error-bounded methods, AC-GC achieves $15.1\times$ compression with an average accuracy change of $0.1\%$ on text and image datasets. AC-GC functions on any model composed of the layers analyzed and, by avoiding compression rate search, reduces overall training time by $4.6\times$ over SuccessiveHalving.

## 1 Introduction

Stochastic Gradient Descent (SGD) has proven efficient and effective for optimizing Deep and Convolutional Neural Networks (DNNs and CNNs). However, due to deeper and automatically generated networks [20, 22, 44, 52], improvement of accuracy has caused a rapid increase in training memory requirements, which are dominated by the temporary storage of activations between the forward and backward pass of the back-propagation algorithm [45, 48]. Reducing memory consumption leads to faster training and, thus, more effective research of DNN models and applications. However, doing this by decreasing the batch size has many drawbacks. On parallel processors, such as GPUs, a small batch size can lead to poor compute saturation, and reduced training throughput [47]. Smaller batch sizes also introduce errors that impact convergence and accuracy [18]. Over 50GB of memory is required to train some networks, e.g., GPIPE [22].

Many works have examined reducing activation storage overheads. Lossy compression of activations in memory can reduce memory footprint without network modifications [6, 14, 25, 27]. Error bounded lossy compression (EBC) [27] has bounded activation error, however, it uses an empirical study to select an error target. Activations can also be offloaded to an external memory (e.g. CPU DRAM), using either an uncompressed link [31, 45] or compressed link [14, 46]. Activation compression and offloading have performance overheads from 5% to 60% [7, 14, 25, 27, 45]. Reduced precision

35th Conference on Neural Information Processing Systems (NeurIPS 2021).

training has the side effect of reducing activation size [9, 51, 57]. Finally, restructuring networks to be reversible [17] or efficient scheduling of network layers [8] can reduce memory use.

Prior lossy and reduced precision approaches [6, 9, 14, 25, 51, 57] utilize automated searches or hand-tuning to determine compression rates, which increase training time and have the potential to select poor compression/accuracy trade-offs. Ideally, an activation compression method has high compression and minimal decrease in trained accuracy. Tuning to achieve this is prohibitively expensive. For instance, selecting a fixed-point integer (fixpoint) compression rate for ImageNet/ResNet50 using Grid Search uses 16 training runs (Figure 1). Using Successive-Halving [26] can decrease training time, however even with aggressive resource allocations (e.g. 24 GPU-days, Figure 1) total training time is still high. With low resource settings, methods such as SuccessiveHalving [26] and Hyperband [36] allocate little time to some configurations, increasing the likelihood that compression

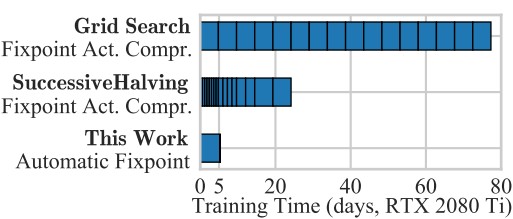

Figure 1: Activation compression rate search cost for ImageNet/ResNet50 [20]. Each box indicates a different compression from 1- to 16-bit fixpoint.

artifacts [6, 25] can be missed, resulting in poor accuracy. Additionally, when tuning hyperparameters, lossy compression makes it difficult to determine the cause of degraded accuracy. Finally, prior compression methods have an unknown impact on convergence behavior.

In this work, we present a framework for lossy Activation Compression with Guaranteed Convergence (AC-GC). To our knowledge, our work is the first to prove convergence bounds on SGD with activation compression. AC-GC involves allowing an increase in the bound on the expected loss, which we trade-off for increased compression. We formulate this as a constrained optimization problem: maximizing compression subject to a bounded increase in loss. Doing so allows using a single hyperparameter to correlate convergence bounds with the activation error, which creates compression methods that are iteration, network, and dataset agnostic. Having convergence bounds known *a priori* allows a user to set a tolerable error rate *before training*, avoiding compression rate search cost entirely. Our contributions:

- We prove convergence bounds on SGD under error bounded activation compression with weak assumptions on convexity.

- We express activation compression and convergence as a constrained optimization problem and analyze the activation error tolerance of common DNN layers within this framework.

- We combine these error bounds with fixpoint, image, and error bounded compression, to create methods with guaranteed convergence and a compression/accuracy trade-off known prior to training.

## 2  Preliminaries

DNNs are commonly used on problems involving a sum, for instance, minimize the total error on a set of training images. The loss $\mathcal{L}$ for such a problem takes the form

$$\mathcal{L}(\theta) = \sum_n f(\theta, X_n) \tag{1}$$

where $f$ represents the loss of one example input $X_n$ with weights $\theta$.

Stochastic Gradient Descent (SGD) is typically used to optimize these finite sums, using the iteration

$$\theta^{(t+1)} = \theta^{(t)} - \alpha \nabla_\theta f(\theta^{(t)}, X_{n_t}) \tag{2}$$

where $\alpha$ is the learning rate, $t$ is the iteration, $\nabla_\theta f$ represents the gradient of $f$ with respect to $\theta$, and $n_t$ is a randomly chosen training example index from the distribution over $n$ such that $\mathbb{E}[\nabla_\theta f(\theta^{(t)}, X_{n_t})] = \nabla_\theta \mathcal{L}(\theta^{(t)})$.

Figure 2a shows the computation graph for the back-propagation [48] algorithm for a single DNN layer without compression. Back-propagation is often used as it allows efficient calculation of

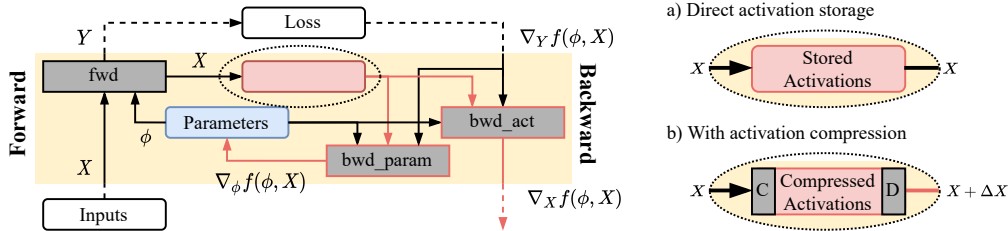

Figure 2: Computation graph for training of a DNN layer. Activations are a) stored between the forward (left) and backward pass (right) or b) compressed (C) in the forward pass, and decompressed (D) in the backward pass. Red indicates paths potentially affected by compression errors.

parameter gradients at the expense of storing activations [48]. A DNN layer is any linear or non-linear function, e.g., a convolution or ReLU activation. The functions `fwd`, `bwd_param`, `bwd_act` are algorithmic implementations of the layer function and the gradients w.r.t. $\theta$ and $X$. In the forward pass, each layer calculates an output activation $Y = \texttt{fwd}(X)$ which is fed to subsequent layers. These activations are temporarily stored after use to avoid a performance penalty from recalculating them in the backward pass. To our knowledge, all frameworks opt to store activations [42, 53]. In the backward pass, parameter gradients and activation gradients are calculated using `bwd_param` and `bwd_act`. Parameter gradients are used to update the parameters (Eqn. (2)), and activation gradients are sent downward to the next layer. Depending on the layer type and its derivatives, the `bwd_param` and `bwd_act` functions may require activations to be stored. For example, it is computationally efficient to store the input $X$ for convolution layers [42, 53]. The many layer types place a diverse set of constraints on the activation storage.

Activation compression addresses one of the most significant contributors to memory consumption in DNNs. In the forward pass, an activation can be lazily compressed after its last usage (C, Figure 2b). Eagerly compressing activations would require storing both a compressed and uncompressed copy until its last use. In the backward pass, activations are decompressed before their first use (D, Figure 2b). The backward pass begins only after the forward pass is completed for all layers, resulting in a large reuse distance for stored activations. Compression can thus be performed off the critical path in parallel with compute, with low performance overheads from 4%-30% [7, 25], provided that sufficient resources are available.

We denote the uncompressed activation as $X = (x_{nchw}) \in \mathbb{R}^{N \times C \times H \times W}$, where $N$, $C$, $H$, and $W$ represent the batch size, channel, height and width, respectively. In the uncompressed backward pass, gradients are calculated from the saved activations, parameters, and gradients from the upward layer (Figure 2 with a); we write this as $\nabla_\theta f(\theta, X) = \texttt{bwd\_param}(X, \theta, \nabla_Y f(\theta, X))$.

Lossy compression involves discarding some information of the activation to increase the compression rate. In information theory, this is referred to as the rate-distortion trade-off. In our model, the rate refers to the activation error, and the distortion refers to resulting impacts on gradient error and thus accuracy after training. We model lossy compression between the forward and backward pass as an independent perturbation on each value in the activation, $\Delta X \in \mathbb{R}^{N \times C \times H \times W}$. Thus, in the compressed backward pass, the perturbed activation is $X + \Delta X$. We denote the approximate gradient resulting from lossy error as $\hat{\nabla}_\theta f(\theta, X) := \texttt{bwd\_param}(X + \Delta X, \theta, \nabla_Y f(\theta, X))$, and the corresponding gradient error as $\Delta \nabla_\theta f(\theta, X) := \hat{\nabla}_\theta f(\theta, X) - \nabla_\theta f(\theta, X)$.

## 3 Guaranteed Convergence

This section details how gradient error $\Delta \nabla_\theta f(\theta, X)$ impacts convergence of SGD. Following this, the lossy compression error $\Delta X$ can be expressed in terms of the gradient error, and the compression rate for many methods can be determined. For example, the bitwidth $b$ of fixpoint compression of an activation with range $(-1, 1)$ is

$$b \geq -\log_2 |\Delta x_{nchw}| + \dots \tag{3}$$

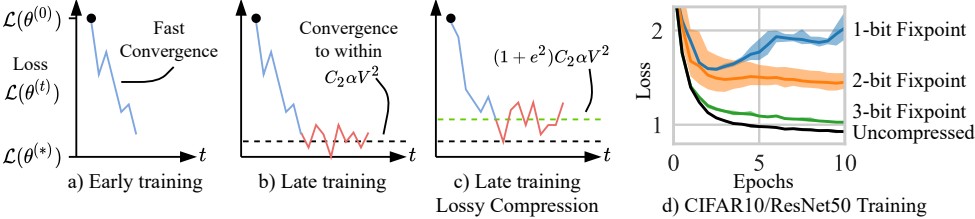

Figure 3: SGD convergence behavior. a) and b) without compression, c) this work, and d) ResNet50 training with $\alpha = 0.25$ and fixpoint activation compression (average over five runs, shaded indicates the minimum and maximum training loss).

where the remaining terms are constants determined by the rounding mode, sign, etc. Any compression method with bounded activation error for a given rate can be combined with the error bounds from this work to guarantee convergence (Sections 3.2 and 4).

## 3.1 SGD Convergence

We will briefly summarize uncompressed convergence of SGD from Karimi et al. [28]. Consider training using $t$ iterations of SGD with loss $\mathcal{L}(\theta^{(t)})$, with a constant learning rate $\alpha$ and initial point $\theta^{(0)}$. We assume that $\mathcal{L}$ has an optimal point $\theta^{(*)}$ and satisfies $\mathbb{E}[\|\nabla_\theta f(\theta, X_{n_t})\|^2] \leq V^2$ for all $\theta$ and some $V^2$. We refer to $V^2$ as the variance. With some assumptions and problem-defined constants $C_1$ and $C_2$ (Appendix A), Karimi et al. [28] demonstrate that the expected error at iteration $t$ is

$$\mathbb{E}[\mathcal{L}(\theta^{(t)}) - \mathcal{L}(\theta^{(*)})] \leq (1 - C_1\alpha)^t(\mathcal{L}(\theta^{(0)}) - \mathcal{L}(\theta^{(*)})) + C_2\alpha V^2 \tag{4}$$

Initially when training, fast convergence occurs as $(1 - C_1\alpha)^t$ approaches zero (Figure 3a). Later in training, $C_2\alpha V^2$ dominates, resulting in an approximately constant expected error (Figure 3b). The key observation of this result is that the final loss scales with gradient variance, $\mathbb{E}[\mathcal{L}(\theta^{(\infty)})] \propto V^2$.

Many DNN classes fall under this progress bound as it uses relatively weak assumptions and does not require a convex $f$. DNNs using ReLU activations are Lipschitz continuous [55]. Furthermore, those with an L2 loss are piecewise strongly convex, which implies that the required Polyak-Łojasiewic condition is satisfied locally [40]. Networks this does not apply to could use another progress bound [3, 28, 41].

## 3.2 Compressed Convergence

Lossy compression trades accuracy for compression. This trade-off can be empirically observed with fixpoint compression on CIFAR10/ResNet50 (Figure 3d and Section 6). Our method functions by *allowing* the loss to increase by some multiplicative error $(1 + e^2)$, where $e^2 \geq 0$, and determining how much compression can be extracted from the change (Figure 3c). The error $1 + e^2$ is chosen such that the compressed loss converges to the uncompressed loss as $e^2 \to 0$.

Our process for translating the loss bound into maximum activation errors is illustrated in Figure 4a. This section outlines how the loss can be bounded by bounding the gradient error or using an intermediate bounding function $D(\Delta X)$ to simplify the problem. We follow this in Section 4 by deriving an activation error $\Delta X^{(*)}$ for each network layer that satisfies this bound. $e^2$ becomes the sole hyperparameter in our method, and selection determines the maximum increase in loss and the compression rate.

Compressed convergence with increased loss can be viewed as an uncompressed problem with an increased gradient variance bound, $(1 + e^2)V^2$. To determine the activation error which satisfies this gradient variance, we must express the variance in terms of a bound on the gradient error $\|\Delta\nabla_\theta f(\theta, X)\|^2$. Theorem 1 demonstrates that a maximum gradient error of $e^2V^2$ satisfies the $(1 + e^2)$ loss bound. There may be multiple regions where the gradient error is below $e^2V^2$ (Figure 4b), which would require iterative solvers to determine suitable activation errors. To apply our technique in practice we introduce a convex function $D(\Delta X) \geq \|\Delta\nabla_\theta f(\theta, X)\|^2$, which provides a flexible

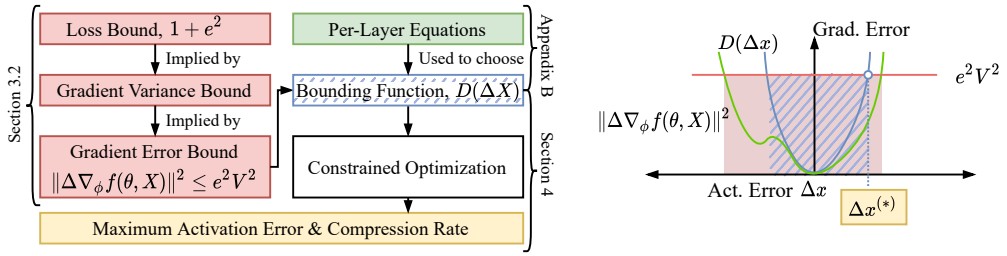

a) Flowchart of Derivations

b) 1D Bounding Function Example

Figure 4: a) Flowchart of derivations for obtaining the maximum activation error $\Delta X^{(*)}$ as a function of the loss bound $1 + e^2$. b) 1D Example for the relationship between the error bound, bounding function, and gradient error for an activation error $\Delta x$. Shaded area satisfies the loss bound, and the hatched area satisfies $D(\Delta x) \leq e^2 V^2$.

proxy for the gradient error (Figure 4b). Using $D(\Delta X)$ allows defining the problem so that it has a unique solution. The gradient variance bound, bounding function, and gradient error are related in Theorem 1.

**Theorem 1.** *Given $f$ which obeys (4), and a convex function $D(\Delta X)$ which bounds the gradient error from above for all $X$, $\theta$, and $\Delta X$:*

$$\|\Delta \nabla_\theta f(\theta, X)\|^2 \leq D(\Delta X) \tag{5}$$

*then any activation error $\Delta X^{(*)}$ where $D(\Delta X^{(*)}) \leq e^2 V^2$ satisfies*

$$\mathbb{E}[\|\hat{\nabla}_\theta f(\theta, X_{n_t})\|^2] \leq (1 + e^2) V^2 \tag{6}$$

*assuming that $\mathbb{E}[\|\Delta \nabla_\theta f(\theta, X_{n_t})\|] = 0$ for all $\theta$, with positive value $e^2$, and variance $V^2$ satisfying $\mathbb{E}[\|\nabla_\theta f(\theta, X_{n_t})\|^2] \leq V^2$. All expectations are taken over the training examples $n_t$.*

*Proof: See Appendix A.*

Figure 4b demonstrates the relationship between the various bounds, as well as the motivation for the bounding function. The gradient error is derived from per-layer equations, and in many cases is highly non-convex. Due to this, local minima can cause difficulties when optimizing with the constraint $\|\Delta \nabla_\theta f(\theta, X)\|^2 \leq e^2 V^2$ (Figure 4b). $D(\Delta X)$ is defined to be convex, making the region defined by the constraint a closed region (hatched, Figure 4b). Although the original constraint could tolerate a higher activation error (and thus compression), using a bounding function provides favorable conditions for obtaining a closed-form solution for the activation error.

## 4 Framework for Activation Compression

For our evaluation of AC-GC, we develop activation error bounds for various commonly used DNN layers and apply them to several recent networks. This section summarizes bounds for common layers, and derivations and additional layer bounds are provided in Appendix B. Calculating compression error from the gradient error bound $e^2 V^2$ requires expressing and solving for the compression/convergence trade-off. We tackle this by formulating the trade-off as a constrained optimization problem: maximizing the compression subject to bounded gradient error. The problem must be solved once per layer type and can be formally defined as

$$\Delta X^* = \underset{\Delta X}{\operatorname{argmax}} B(\Delta X) \quad \text{s.t.} \quad D(\Delta X) = e^2 V^2 \tag{7}$$

$$\text{where} \quad D(\Delta X) \geq \|\Delta \nabla_\theta f(\theta, X)\|^2 \tag{8}$$

where $B(\Delta X)$ is a continuous convex function that measures compression rate as a function of the activation error $\Delta X \in \mathbb{R}^{N \times C \times H \times W}$. A closed-form solution can be found for many systems of this type using the method of Lagrange multipliers. The constraint $D(\Delta X) \leq e^2 V^2$ defines a convex region of potential activation errors where the convergence constraint is satisfied (hatched, Figure 4b).

However, as both the $B(\Delta X)$ and $D(\Delta X)$ functions are convex, the maximum value must occur along the boundary, hence the equality constraint $D(\Delta X) = e^2 V^2$ in (7) ($\Delta x^{(*)}$, Figure 4b). The convexity of $D(\Delta X)$ also implies that $\Delta X^{(*)}$ is the maximum activation error, i.e., any error $(\Delta X)^2 \le (\Delta X^{(*)})^2$ also satisfies the variance bound (6).

Many compression methods use a variation of fixpoint. Hence, we select $B$ to measure the number of bits removed from the activation when compressed with reduced precision fixpoint (9). Rounding mode and sign are constant factors that do not affect the result, however, we ignore clipping. The target compression method loosely influences the objective, hence, non-fixpoint methods may fare better with another error objective.

$$B(\Delta X) := \sum_{n,c,h,w}^{N,C,H,W} \log |\Delta x_{nchw}| \tag{9}$$

To derive AC-GC error bounds for DNN layers not presented in this work, one can:

1. Derive $\|\Delta \nabla_\theta f(\theta, X)\|^2$ for the layer type
2. Choose a suitable convex bounding function, $D(\Delta X) \ge \|\Delta \nabla_\theta f(\theta, X)\|^2$
3. Obtain the maximum error $\Delta X^{(*)}$ by solving (7) using the method of Lagrange multipliers

For the sake of brevity, we will summarize the notation, assumptions, and AC-GC error bounds for fully connected, convolution, and batch normalization layers (Table 1). We aim to locate closed-form solutions with low computation overheads, although tighter bounding functions likely exist. Henceforth we omit arguments of $f$ and use the following definitions: Batch size $N$, input channels $C$, output channels $K$, input and output activations $X$ and $Y$, and compression error $\Delta X$.

**A) Fully Connected:** Table 1A relates the error for guaranteed convergence with compression error for a fully connected layer, with weights $\theta = (\theta_{kc}) \in \mathbb{R}^{K \times C}$, input activation $X = (x_{nc}) \in \mathbb{R}^{N \times C}$, and output activation gradient $\nabla_Y f = (\partial f / \partial y_{nk}) \in \mathbb{R}^{N \times K}$. As the error bound ($e^2 V^2 / 2$) decreases, the compression must decrease to compensate. All activations for a fully connected layer have the same error tolerance.

**B) Convolution:** Convolution with no padding follows a similar trend to linear layers, with the addition of stride $T$, a filter size of $R \times S$, and increased dimensions $X = (x_{nchw}) \in \mathbb{R}^{N \times C \times H \times W}$ and $\nabla_Y f = (\partial f / \partial y_{nkhw}) \in \mathbb{R}^{N \times K \times H \times W}$. We assume an average usage of activations due to stride, as uneven usage leads to uneven compression, which would require tracking per-element compression rates. Comparing linear and convolution reveals that convolutions have a lower error tolerance due to the increased number of activations ($HW$) and weights ($RS$).

To fully cover cases encountered in CNNs, we also derive error bounds for cases where activation errors affect multiple convolution layers (e.g., in ResNets [20]).

**C) Batch Normalization:** Batch normalization [24] re-normalizes the activation from per-channel standard deviation $\sigma \in \mathbb{R}^C$ to a learned $\gamma \in \mathbb{R}^C$. There is a different dependence on activation error from convolution and linear layers. Instead of causing errors in the parameter gradient $\nabla_\gamma f$ exclusively, activation error propagates to the activation gradient $\nabla_X f$ and then to all subsequent layers in the network. To avoid bounding all weights in the network, we isolate the layer and bound the convergence of the batch normalization activations using

$$\|\Delta \nabla_X f\|^2 \le e^2 V^2 \tag{10}$$

Arriving at a closed-form solution further requires a bound on the parameter gradient error within the layer using positive values $(g_c^2) \ge (\hat{\nabla}_\gamma f)^2$ (Appendix B). Although not observed for the networks in this work, as network parameter gradients are not directly bounded, the convergence bounds on networks with batch normalization may be violated.

**Layer Normalization:** This layer type is similar to batch normalization and requires a similar set of assumptions and derivations (Appendix B).

**ReLU, Dropout, Max Pooling, and Summation:** Summation does not require storage of any activation, and the remaining layers (Dropout, Max Pooling, and ReLU) only require a bitmask to calculate their respective gradients. For instance, ReLU requires the storage of the bitmask $X \ge 0$ [14, 25], and Max Pooling requires a bitmask of the locations of maximal values. As these layers have an efficient lossless high compression method available, we do not analyze them.

Table 1: Guaranteed convergence equations for common network layers. See Appendices for full derivations and assumptions. Empty sums are over all indicies, i.e. $\sum := \sum_{n,c,h,w}^{N,C,H,W}$. $M := NHW$

| QUANTITY | A) FULLY CONNECTED | B) CONVOLUTION | C) BATCH NORMALIZATION |
|---|---|---|---|
| $D(\Delta X) :=$ | $\|\nabla_Y f\|^2 \|\Delta X\|^2$ | $\dfrac{RS}{T^2} \|\nabla_Y f\|^2 \|\Delta X\|^2$ | $\sum \dfrac{2\gamma_c^2 g_c^2}{M^2 \sigma_c^4} \Delta x_{nchw}^2$ |
| $\Delta x_{nchw}^{(*)2} =$ | $\dfrac{e^2 V^2}{2NC\|\nabla_Y f\|^2}$ | $\dfrac{e^2 V^2 T^2}{2RSMC\|\nabla_Y f\|^2}$ | $\dfrac{e^2 V^2 M \sigma_c^4}{C \gamma_c^2 g_c^2}$ |

## 5 Practical Automatic Lossy Compression

Using the AC-GC error bounds, we create convergence bounded compression methods with automatic compression rates, collectively referred to as AutoX. The first two methods (AutoQuant and AutoCuSZ) adapt scaled fixpoint [14] and error-bounded compression [27], which have bounded errors for a given compression rate. The errors for these methods are unbiased provided that unbiased rounding to fixpoint is used [5, 27]. The third method (AutoJPEG) uses lossy JPEG compression [14]. We bound JPEG error using an empirical error-compression relationship using activations sampled from uncompressed training of CIFAR10/ResNet50 [20]. Samples are used offline with JPEG compression to establish the compression-error relationship. This empirical compression-error relationship is used to calculate the JPEG compression levels, which approximately satisfy the error bounds in Table 1. We chain quantization and JPEG with lossless Zero Value Compression [46] to compress sparse activations better, creating AutoQuantZ and AutoJPEGZ.

As all AutoX used some form of fixpoint, we can express activation error in terms of bits. For any layer type, the relationship between bitwidth $b$ and the convergence bound can be described as

$$b \geq -\log_2 |\Delta x_{nchw}| + ... = -\log_2 |e| - \log_2 \|\nabla_\theta f\| + ... \tag{11}$$

Using the results from Table 1, it can be seen that the bitwidth scales additively with the batch size as $+\log_2(N)$ for convolution layers, and $-\log_2(N)$ for normalization layers.

Two issues with using AC-GC in a compression method are: 1) the various norms required are computationally expensive, and 2) the formulation assumes exact gradient information is available during the forward pass. We address both issues by statistically estimating activation error bounds. Instead of evaluating at every iteration, statistics are calculated at the end of every *recalculation interval*, specified in iterations. In the forward pass, a *summary* (mean or maximum) of the last ten recalculations is used when calculating the errors. This also allows approximating $V^2 \approx \|\nabla_\theta f\|^2$. Although some quantities are available in the forward pass, we estimate all of the activations, parameters, and gradients to avoid performing norm calculations at every iteration. Despite norms being estimated, we do not observe that the convergence bound (6) is violated for any network examined. A few training iterations can be used to verify correctness of the norm estimates (Figure 5c).

## 6 Evaluation

We examine activation compression by modifying the Chainer framework [53] to compress and decompress activations during training. We measure compression rates every 100 iterations, and otherwise perform paired compression/decompression to maintain the highest performance for our experiments. We focus our analysis on CNNs with image and text datasets, as they have large activation memory requirements, but avoid the largest networks [22, 52] due to limited resources. We create a performance implementation based off Chen et al. [7] to measure throughput.

For ImageNet [11], CIFAR10 [2] and Div2K [1], we use SGD and 0.9 momentum for VGG16 [50], ResNets (RN18 and RN50) [20], Wide ResNet (WRN) [59], and VDSR [29]. IMDB [39] and Text Copy [4] are trained using ADAM with CNN [53], RNN [53], and transformer heads [54]. All image datasets are augmented with random sizing, flip, and crop, as well as whitening and PCA for ImageNet [30], and $8 \times 8$ cutout for CIFAR10 [12]. Learning rates, batch sizes, and epochs are 0.05, 128, 300 (CIFAR10, [49]), 0.1, 64, 105 (ImageNet, [58]), 0.1, 32, 110 (Div2K, grid search), 2.0, 64, 100 (Text Copy, [4]), and 0.001, 64, 20 (IMDB, [53]).

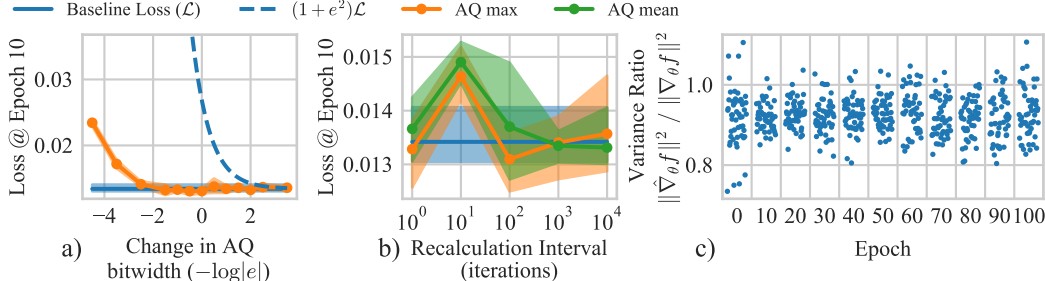

Figure 5: a-b) Average loss over the 10th epoch on MNIST/LeNet, where shaded regions indicate the min/max loss over five runs. a) Loss as function of $e$. Under compression, the empirical loss (AQ max: AutoQuant with a maximum summary) falls below the theoretical loss bound $(1 + e^2)\mathcal{L}$. b) Loss where statistics are calculated every *recalculation interval* iterations. The max or mean of the last ten intervals is used to calculate the bitwidth. c) Ratio of compressed to uncompressed gradients for the first convolution for ImageNet/ResNet18, where the compressed gradient is used to update the parameters. Clusters indicate the first 50 iterations of every ten epochs.

All forward and backward pass calculations use floating-point, using activations that have been compressed and decompressed between the two passes. Baseline refers to uncompressed training, i.e., 32-bit floating-point activations. GridQuantZ uses the same implementation of AutoQuantZ, however, it uses grid search over eight bit-widths of $2, 3, 4, 6, 8, 10, 12$ and $16$ bits, and then chooses the lowest with accuracy within 0.1% of the baseline. These grid points were selected to give good coverage of low and medium bit-widths, and are approximately logarithmic-spaced. SuccessiveHalving [26] and Hyperband [36] produce similar accuracy/compression to grid search but take less time. Unless otherwise stated, all experiments use $e^2 = 0.5$, parameter estimates from the mean of a ten entry window, and a recalculation interval of 100 iterations. A value of $e^2 = 0.5$ allows a small increase in loss over the baseline ($+50\%$), but other values could be chosen depending on the desired error tolerance. Appendices C and D contain additional detail on hyperparameters and implementations.

## 6.1 Parameter Sensitivity

We isolate the impacts of parameter estimation and $e$ selection by training LeNet [33] on MNIST [32], and RN18 [20] on ImageNet [11]. We train MNIST to convergence by using SGD with no momentum, a learning rate of 0.001, a batch size of 64, and 10 epochs.

The effect of $e$, recalculation interval, and summary method are evaluated by training MNIST/Lenet with different configurations of AutoQuant (AQ) (Figures 5a and 5b). Decreasing $e$ (which increases bitwidth) causes the loss to increase, however, the average loss does not violate the bound in Theorem 1 (Figure 5a). In general, all networks examined with $e^2 = 0.5$ have loss changes below 2% and validation score changes below 0.5% (Sections 6.2 and 6.3), which demonstrates that AC-GC error bounds are not violated. Both interval and summary method have a minimal impact on the training loss for MNIST (Figure 5b). As there is an insignificant change in loss, we use the mean for AC-GC as it has higher compression.

We evaluate ImageNet/ResNet18 using a dual training approach to ensure the correctness of parameter estimation (Figure 5c). This involves training using AutoQuant, while evaluating the *true* weight gradients $\nabla_\theta f$ offline and comparing against their compressed counterpart $\hat{\nabla}_\theta f$. The ratio is $\approx 1$ for the duration of training, and does not violate the bound for $e^2 = 0.5$ (i.e., $\leq 1.5$). The mean ratio is less than one, likely due to decreased activation variance from compression to a discrete set of values. We observe similar behavior for the other layers in the network (not shown).

## 6.2 CIFAR10, Div2K and IMDB

We compare the AutoX methods with fixpoint grid search (GridQuantZ), and with prior works on lossy JPEG compression [14] (Table 2). Grid search requires oracle knowledge of the baseline, and $8\times$ the training iterations of any other method in Table 2. Compared to GridQuantZ, AutoQuantZ uses a single run of training, and provides a similar compression rate of $7.5\times$. AutoCuSZ has a

Table 2: Test/validation score and compression rate (bracketed) for fixpoint with grid search (GridQuantZ), AutoX methods, and JPEG-ACT (optL5H from [14]), averaged over 3 runs (ImageNet) or 5 runs (remainder). The highest accuracy and compression are bolded. Trained using 900 GPU-days (RTX 2080 Ti). N/A indicates either not run in [14], or lack of spatial activations.

| Model | Base | Auto QuantZ | Auto CuSZ | Auto JPEGZ | JPEG-ACT[14] | Grid QuantZ |
|---|---|---|---|---|---|---|
| **CIFAR10** % Top-1 Test Accuracy | | | | | | |
| VGG | 93.6 | **93.5** ( 7.4×) | **93.5** ( 9.4×) | 92.9 (**12.5**×) | 92.4 (11.9×) | **93.5** ( 6.3×) |
| RN50 | 94.9 | **95.0** ( 4.2×) | 94.7 (**15.5**×) | 94.3 ( 9.2×) | 94.4 ( 7.5×) | **95.0** ( 5.7×) |
| WRN | 95.8 | 95.9 ( 6.5×) | 95.8 (**14.6**×) | 95.3 (11.7×) | 94.2 (10.9×) | **96.0** ( 7.6×) |
| **Div2K** Best Val. PSNR | | | | | | |
| VDSR | 36.1 | **36.1** ( 5.1×) | 35.8 (**25.2**×) | **36.1** ( 7.9×) | 35.4 ( 9.1×) | 36.0 ( 6.7×) |
| **IMDB** % Best Val. Accuracy | | | | | | |
| CNN | 61.4 | 61.6 (12.2×) | **61.8** (**19.3**×) | 61.4 (11.2×) | N/A | 61.7 (16.5×) |
| LSTM | 60.3 | 60.1 (10.0×) | **60.9** ( 8.8×) | N/A | N/A | 60.4 (**14.7**×) |
| **Text Copy** % Best Test Accuracy | | | | | | |
| TRANS | 98.8 | 98.6 ( 7.1×) | 98.3 (**12.7**×) | N/A | N/A | **98.9** ( 5.1×) |
| **ImageNet** % Top-1 Center Crop Val. Accuracy | | | | | | |
| RN18 | 68.6 | **68.5** ( 4.2×) | 68.1 ( 6.8×) | 68.1 ( 8.1×) | 67.3 ( 7.2×) | **68.5** ( 2.9×) |
| RN50 | 72.3 | **72.7** ( 4.8×) | 72.5 (**10.1**×) | 71.5 ( 8.5×) | 71.6 ( 5.9×) | 72.5 ( 4.9×) |
| **Average** %-point Change and Compression ratio | | | | | | |
| ALL | 0 | +0.0 ( 7.5×) | −0.1 (**15.1**×) | −0.6 (10.5×) | −1.0 ( 7.4×) | +**0.1** ( 7.8×) |

compression 2.0× higher than JPEG-ACT, while maintaining accuracy to within 0.1 of the baseline on average. With a suitable error bound, CuSZ can extract significant compression from zeros and spatial information. On non-spatial data and non-image datasets (Text Copy and IMDB, Table 2) we observe that AutoCuSZ extracts similarly high compression with little accuracy change. Finally, AutoJPEGZ vs. JPEG-ACT demonstrates that using AC-GC error bounds gives higher accuracy and compression than using heuristics to select JPEG hyperparameters. In general, we find that using $e^2 > 0.5$ decreases accuracy, leaving little reason to modify it.

## 6.3 ImageNet

On ImageNet training with ResNets (Table 2), the AutoX methods obtain a high accuracy. Our ImageNet accuracies are lower than other works as we do not use random scaling (which improves performance), and we report 1-crop accuracy. Our 10-crop accuracy for the ResNet50 baseline is 75.2%. Reduced compression rates on ResNet18 vs. ResNet50 are due to a lower sparsity in ReLU activations, which we hypothesize is caused by the different bottleneck structures of the two networks [20]. The higher-than-baseline accuracy of AutoQuantZ (Table 2) is caused by a large standard deviation for the ImageNet baseline (±0.21). On ImageNet, AutoCuSZ gives a high compression in exchange for a small decrease in accuracy, 0.15%-points. AutoQuantZ provides high accuracy, at a moderate compression rate of 0.7×, with a 1.2%-point better accuracy than JPEG-ACT. However, the primary advantage of AutoX methods is that the pre-training bound on loss increases.

## 6.4 Overheads

The AutoX methods require error bound calculation (common to all methods) and compression. Our unoptimized AutoQuantZ implementation achieves throughput of 1.66× ($N = 128$) vs. naive swapping to the CPU ($N = 128$), and 0.64× vs. uncompressed training ($N = 32$) (ImageNet/ResNet50). Our primary contribution, AC-GC error bound calculation, uses 0.4% of total training time. This is negligible when compared to compression overhead, e.g., 4% for fixpoint [14, 25], 17% for CuSZ [27], 33% for ActNN-L3 [7], or 13% for hardware accelerated JPEG [14]. Unoptimized AutoQuantZ is 23% slower compared to ActNN-L3 [7]. Compression rate search time is decreased by 4.6× when compared to SuccessiveHalving (Figure 1).

Table 3: Comparison with prior works on CIFAR10 (C10) and ImageNet (IN). $\pm$ indicates standard deviation, if available. Accuracy is presented relative to the baseline accuracy of each work. $^*$ Does not include 2$\times$ memory reduction from recalculating activations, which is orthogonal to this work.

| | AUTO CUSZ | WAGE [57] | BAA$^*$ [6] | JPEG- ACT[14] | ULP [51] | EBC [27] | ACTNN -L3[7] |
|---|---|---|---|---|---|---|---|
| **DATASET** | IN | C10 | IN | IN | IN | IN | IN |
| **MODEL** | RN50 | VGG16 | RN152 | RN50 | RN50 | RN50 | RN152 |
| **METHOD** | AUTO -CUSZ | 8-BIT | 4-BIT +RECALC. | JPEG +ZVC | 4-BIT | CUSZ | 2-BIT MIX. PREC. |
| **ACC. (%)** | $-0.2\pm0.2$ | $-0.3$ | $-0.5$ | $-0.1$ | $-0.3$ | $-0.9$ | $-0.2$ |
| **COMPR.** | 10.1$\times$ | 4$\times$ | 8$\times$ | 5.9$\times$ | 8$\times$ | 11.0$\times$ | 12$\times$ |

## 7   Related Works

We compare AutoCuSZ with the most recent works in activation compression [6, 14, 25, 27] and reduced precision training [51, 57] (Table 3). AutoCuSZ obtains better accuracy and compression than most works, with the added benefit of not requiring a search over compression rates. BAA [6] presents a technique of recalculating activations similar to Chen et al. [8], which could be combined with any method in Table 3, including the AutoX methods. AutoCuSZ obtains higher compression and accuracy than EBC [27], demonstrating that AC-GC error bounds are better than hand-tuning in this case. WAGE [57] and ULP [51] and reduced precision training with theoretical convergence bounds [35, 37] reduce training compute through reduced precision, however, have lower accuracy due to accumulating errors in both the forward and backward pass. AutoX methods can potentially be combined with low precision training and other techniques to reduce memory requirements further.

ActNN is most similar to this work in that it invokes gradient variance to partition bits among compressed activations [7]. However, ActNN approaches the problem in an opposite manner to this work: it finds the highest accuracy for a given compression rate. The ActNN approach works well when the goal is to fit within a memory budget. Our approach allows for selecting the target loss increase *a priori*, and gaining information on the uncompressed accuracy from a single training run. AC-GC will also converge similarly to the uncompressed case on an unknown model, providing assurances that loss changes are due to model/hyperparameter configuration, not compression. Additionally, ActNN is specific to group-wise fixpoint compression, whereas AC-GC is quick to adapt to any lossy compression method (including group-wise fixpoint). On ImageNet/ResNets, AutoCuSZ and AutoQuantZ obtain half the accuracy change of ActNN-L3, albeit at 0.9$\times$ and 0.4$\times$ the compression, respectively.

Works targeting inference, such as precision reduction [10, 15, 23, 34], compression [13, 19, 38, 56], and sparsification [16, 43] increase memory requirements due to tracking additional state. Other works that directly address memory can be grouped into scheduling [8, 25], offloading [31, 45], and restructuring [17, 21]. Generally, these come with a performance or model-flexibility penalty. Any non-lossy method is partially orthogonal to compression and can be potentially combined with AC-GC for increased memory reduction.

## 8   Conclusions

The AC-GC automatic compression methods described in this work provide high compression rates with error trade-offs known before training. Avoiding compression rate search comes at a computational overhead of only 0.4%, many times less than tuning techniques. Provided that the assumptions of AC-GC hold, it can be further combined with any lossy compression method, layer, or network to guarantee convergence with high activation compression. Although convergence is theoretically guaranteed, some factors could impact convergence, such as floating-point errors. By lowering training costs, this framework allows for training larger models and faster exploration of the machine learning field's landscape. Detailed derivations and implementations can be found in the supplemental material. Code is available `https://github.com/rdevans0/acgc`.

## Funding Transparency Statement

This research was funded in part by the Computing Hardware for Emerging Intelligent Sensory Applications (COHESA) project financed under the National Sciences and Engineering Research Council of Canada (NSERC); grant number NETGP485577-15.

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
