# Supplemental Material for
# AC-GC: Lossy Activation Compression with Guaranteed Convergence

**R. David Evans**
Dept. of Electrical and Computer Engineering
University of British Columbia
Vancouver, BC V6T 1Z4
rdevans@ece.ubc.ca

**Tor M. Aamodt**
Dept. of Electrical and Computer Engineering
University of British Columbia
Vancouver, BC V6T 1Z4
aamodt@ece.ubc.ca

The appendices of this supplemental material are focused on providing detailed proofs (Appendix A), per-layer derivations for activation errors (Appendix B), algorithm and implementation details (Appendix C), datasets and hyperparameters (Appendix D), extended experimental data (Appendix E) and additional experiments (Appendix F) to accompany the main paper.

A code example and trained models are available for CIFAR10/ResNet50 by accessing https://github.com/rdevans0/acgc.

## Appendix A   Proof of Convergence Bounds

This section details the assumptions and proof of Theorem 1 from the main text.

**Assumptions for Theorem 1**   The uncompressed SGD bound (4) from Karimi et al. [8] assumes the following for the loss $\mathcal{L}$:

- has a non-empty solution set $\theta^{(*)}$
- has a gradient which is $L$-Lipschitz, for some $L$
- has a gradients satisfying

$$\mathbb{E}[\|\nabla_\theta f(\theta, X_{n_t})\|^2] \leq V^2 \tag{12}$$

  for all $\theta$ and some $V^2$
- satisfies the Polyak-Łojasiewic (PL) condition for all $\theta$ and some $\eta$:

$$\frac{1}{2}\|\nabla_\theta \mathcal{L}(\theta)\|^2 \geq \eta(\mathcal{L}(\theta) - \mathcal{L}(\theta^{(*)})) \tag{13}$$

The full form of the progress bound (4) after $t$ iterations of SGD from Karimi et al. [26] is

$$\mathbb{E}[\mathcal{L}(\theta^{(t)}) - \mathcal{L}(\theta^{(*)})] \leq (1 - 2\eta\alpha)^t(\mathcal{L}(\theta^{(0)}) - \mathcal{L}(\theta^{(*)})) + \frac{L}{4\eta}\alpha V^2 \tag{14}$$

35th Conference on Neural Information Processing Systems (NeurIPS 2021).

which is equivalent to (4) with $C_1 = 2\eta$ and $C_2 = L/(4\eta)$. $L$ and $\eta$ depend on the model being trained and dataset, and are thus problem-dependent constants.

In addition to these assumptions for the uncompressed SGD bound, Theorem 1 requires that the gradient error $\Delta\nabla_\theta f(\theta, X_{n_t})$ is unbiased, i.e.:

- $\mathbb{E}[\Delta\nabla_\theta f(\theta, X_{n_t})] = 0$ for all $\Delta X$ and $\theta$. This is equivalent to $\mathbb{E}[\hat{\nabla}_\theta f(\theta, X_{n_t})] = \mathbb{E}[\nabla_\theta \mathcal{L}(\theta)]$.

**Preliminary on Separation of Norms** Given two, independent random vectors $A = (a_n) \in \mathbb{R}^N$ and $B = (b_n) \in \mathbb{R}^N$, where $\mathbb{E}[b_n] = 0 \;\forall\; n$. The expectation of the norm can be simplified as follows:

$$
\begin{aligned}
\mathbb{E}[\|A + B\|^2] &= \mathbb{E}\left[\sum_n^N (a_n + b_n)^2\right] \\
&= \mathbb{E}\left[\sum_n^N a_n^2 + 2a_n b_n + b_n^2\right] \\
&= \mathbb{E}[\|A\|^2] + 2\sum_n^N \mathbb{E}[a_n]\mathbb{E}[b_n] + \mathbb{E}[\|B\|^2] \\
&= \mathbb{E}[\|A\|^2] + \mathbb{E}[\|B\|^2]
\end{aligned}
\tag{15}
$$

**Proof of Theorem 1**

**Theorem 1.** *Given $f$ which obeys (4), and a convex function $D(\Delta X)$ which bounds the gradient error from above for all $X$, $\theta$, and $\Delta X$; provided that $D(\Delta X) \le e^2 V^2$ the variance of the compressed gradients satisfies*

$$
\mathbb{E}[\|\hat{\nabla}_\theta f(\theta, X_{n_t})\|^2] \le (1 + e^2)V^2
\tag{16}
$$

*Proof.* By definition, the gradient error is bounded above by $D(\Delta X)$ which is bounded by $e^2 V^2$, i.e.

$$
\|\Delta\nabla_\theta f(\theta, X)\|^2 \le D(\Delta X) \le e^2 V^2
\tag{17}
$$

As all quantities are positive, the expected error also cannot exceed the bound,

$$
\mathbb{E}[\|\Delta\nabla_\theta f(\theta, X_{n_t})\|^2] \le e^2 V^2
\tag{18}
$$

where the expectation is taken with respect to $n_t$, over the distribution of examples.

We can use the the separation of norms (15) and the definition of $V^2$ to separate out the variances. This requires that the gradient errors are unbiased, $\mathbb{E}[\Delta\nabla_\theta f(\theta, X_{n_t})] = 0$.

$$
\begin{aligned}
\mathbb{E}[\|\hat{\nabla}_\theta f(\theta, X_{n_t})\|^2] &= \mathbb{E}[\|\nabla_\theta f(\theta, X_{n_t}) + \Delta\nabla_\theta f(\theta, X_{n_t})\|^2] \\
&= \mathbb{E}[\|\nabla_\theta f(\theta, X_{n_t})\|^2] + \mathbb{E}[\|\Delta\nabla_\theta f(\theta, X_{n_t})\|^2] \\
&\le (V^2) + (e^2 V^2) \\
&\le (1 + e^2)V^2
\end{aligned}
\tag{19}
$$

$\square$

Under compression, we use the compressed gradient as an estimate of the true gradient, in a similar way to uncompressed SGD. One could define the compressed variance $\hat{V}^2$ for this problem in a similar manner to (12) as

$$
\mathbb{E}[\|\hat{\nabla}_\theta f(\theta, X_{n_t})\|^2] \le \hat{V}^2
\tag{20}
$$

and then write a similar convergence bound for this problem as

$$
\mathbb{E}[\mathcal{L}(\theta^{(t)}) - \mathcal{L}(\theta^{(*)})] \le (1 - C_1\alpha)^t (\mathcal{L}(\theta^{(0)}) - \mathcal{L}(\theta^{(*)})) + C_2\alpha\hat{V}^2
\tag{21}
$$

where, according to Theorem 1, $\hat{V}^2 = (1 + e^2)V^2$ and thus compressed SGD converges similarly to uncompressed SGD, with a higher gradient variance.

## Appendix B   Derivations for Network Layers

In this appendix, we derive the bounding functions and error bounds for Table 1. The guiding principle of these derivations is to choose bounding functions that have efficient, closed-form solutions. We accomplish this primarily by finding a quadratic (in $\Delta X$) $D$ using the Cauchy-Schwarz inequality.

From this point forward, all gradient values are of $f(\theta, X)$, hence we omit the arguments and write $\nabla_X f$ or $\partial f / \partial x_{nchw}$ for the gradients or derivatives with respect to $X = (x_{nchw})$.

We maximize compression by maximizing the number of bits removed from the compressed activation. The derivative of $B(\Delta X)$ (8) is

$$\frac{\partial B(\Delta X)}{\partial \Delta x_{nchw}} = \frac{1}{\Delta x_{nchw}} \tag{22}$$

### B.1   Fully Connected

For weights $\theta = (\theta_{kc}) \in \mathbb{R}^{K \times C}$, an input activation $X = (x_{nc}) \in \mathbb{R}^{N \times C}$, and output activation gradient $\nabla_Y f = (\partial f / \partial y_{nk}) \in \mathbb{R}^{N \times K}$. The equations describing the fully connected layer forward and backward pass are:

| Shape | Forward | Backward | |
|-------|---------|----------|---|
| $(N, C)$ | $x_{nc}$ : from prev. layer | $\frac{\partial f}{\partial x_{nc}} = \sum_k^K \theta_{kc} \frac{\partial f}{\partial y_{nk}}$ | (23) |
| $(K, C)$ | $\theta_{kc}$ : parameter | $\frac{\partial f}{\partial \theta_{kc}} = \sum_n^N x_{nc} \frac{\partial f}{\partial y_{nk}}$ | (24) |
| $(N, K)$ | $y_{nk} = \sum_c^C \theta_{kc} x_{nc}$ | $\frac{\partial f}{\partial y_{nk}}$ : from next layer | |

The activation gradient (23) is independent of $X$, and thus not affected by activation error. However, the the weight gradient error (24) is affected by activation error. We can calculate the gradient error directly with substitution of $x_{nc} \to x_{nc} + \Delta x_{nc}$ into (24), and bound the error using the Cauchy-Schwarz inequality:

$$\|\Delta \nabla_\theta f\|^2 = \sum_{k,c}^{K,C} \left( \sum_n^N (x_{nc} + \Delta x_{nc}) \frac{\partial f}{\partial y_{nk}} - \sum_n^N (x_{nc}) \frac{\partial f}{\partial y_{nk}} \right)^2 \tag{25}$$

$$= \sum_{k,c}^{K,C} \left( \sum_n^N \Delta x_{nc} \frac{\partial f}{\partial y_{nk}} \right)^2 \tag{26}$$

$$\leq \left( \sum_{n,k}^{N,K} \left( \frac{\partial f}{\partial y_{nk}} \right)^2 \right) \left( \sum_{n,c}^{N,C} \Delta x_{nc}^2 \right) \tag{27}$$

As this is a low order, quadratic bound for $\|\Delta \nabla_\theta f\|^2$, we select it for $D$. In vector form, the bound and its derivatives are

$$D(\Delta X) := \|\nabla_Y f\|^2 \|\Delta X\|^2 \quad (28) \qquad \frac{\partial D(\Delta X)}{\partial \Delta x_{nc}} = 2\|\nabla_Y f\|^2 \Delta x_{nc} \quad (29)$$

The final step is to solve for the maximum activation error $\Delta X^{(*)}$. Substituting $D$ ((28) and (29)) into the system from the method of Lagrange multipliers (30) and solving for $\Delta X^{(*)}$:

$$\frac{\partial B(\Delta X)}{\partial \Delta x_{nc}} = \lambda \frac{\partial D(\Delta X)}{\partial \Delta x_{nc}} \qquad ; \qquad D(\Delta X) = e^2 V^2 \tag{30}$$

$$\frac{1}{\Delta x_{nchw}} = 2\lambda \|\nabla_Y f\|^2 \Delta x_{nc} \tag{31}$$

$$\Delta x_{nc}^2 = \frac{1}{2\lambda \|\nabla_Y f\|^2} \tag{32}$$

Applying the constraint to find $\lambda$:

$$D(\Delta X^{(*)}) = e^2 V^2 \tag{33}$$

$$\|\nabla_Y f\|^2 \sum_{n,c}^{N,C} (\Delta x_{nc}^{(*)})^2 = e^2 V^2 \tag{34}$$

$$\|\nabla_Y f\|^2 \sum_{n,c}^{N,C} \left( \frac{1}{2\lambda \|\nabla_Y f\|^2} \right) = e^2 V^2 \tag{35}$$

$$\lambda = \frac{NC}{2e^2 V^2} \tag{36}$$

resulting in the solution

$$\left( \Delta x_{nc}^{(*)} \right)^2 = \frac{e^2 V^2}{NC \|\nabla_Y f\|^2} \tag{37}$$

## B.2   Simple Convolution

We first begin with a simplified convolution derivation, which will lead to the advanced case. By simple, we do not consider bias, stride, or padding, and assume that error only affects a single convolution layer. We define the input activation $X$ to have dimensions $N \times C \times H \times W$ (batch size, channels, height, width) with a filter size of $R \times S$ and $K$ output channels. For this derivation, we assume the output dimensions are $N \times K \times H \times W$. The equations describing such a layer, in scalar form are:

| Shape | Forward | Backward |
|---|---|---|
| $(N, C, H, W)$ | $x_{nchw}$ : from prev. layer | $\frac{\partial f}{\partial x_{nchw}} = \sum_{k,r,s}^{K,R,S} \theta_{kcrs} \frac{\partial f}{\partial y_{n,k,h-r,w-s}}$ |
| $(K, C, R, S)$ | $\theta_{kcrs}$ : parameter | $\frac{\partial f}{\partial \theta_{kcrs}} = \sum_{n,h,w}^{N,H,W} x_{n,c,h+r,w+s} \frac{\partial f}{\partial y_{nkhw}}$ |
| | | $\qquad\qquad\qquad\qquad\qquad$ (38) |
| $(N, K, H, W)$ | $y_{nkhw} = \sum_{c,r,s}^{C,R,S} \theta_{kcrs} x_{n,c,h+r,w+s}$ | $\frac{\partial f}{\partial y_{nkhw}}$ : from next layer |

Using a similar method as that used to derive the fully connected layer bounding function (28), we can use (38) and the Cauchy-Schwarz inequality to arrive at a suitable $D(\Delta X)$:

$$\|\Delta \nabla_\theta f\|^2 = \sum_{k,c,r,s}^{K,C,R,S} \left( \sum_{n,h,w}^{N,H,W} (x_{n,c,h+r,w+s} + \Delta x_{n,c,h+r,w+s}) \frac{\partial f}{\partial y_{nkhw}} \right.$$

$$\left. - (x_{n,c,h+r,w+s}) \frac{\partial f}{\partial y_{nkhw}} \right)^2 \tag{39}$$

$$= \sum_{k,c,r,s}^{K,C,R,S} \left( \sum_{n,h,w}^{N,H,W} \Delta x_{n,c,h+r,w+s} \frac{\partial f}{\partial y_{nkhw}} \right)^2 \tag{40}$$

$$\leq \sum_{r,s}^{R,S} \left( \sum_{n,c,h,w}^{N,C,H,W} \Delta x_{n,c,h+r,w+s}^2 \right) \left( \sum_{n,k,h,w}^{N,K,H,W} \left( \frac{\partial f}{\partial y_{nkhw}} \right)^2 \right) \qquad (41)$$

Each error $\Delta x_{nchw}$ will be used at most $RS$ times, so we can use $\sum_{r,s}^{R,S}(...) \leq RS(...)$ resulting in:

$$D(\Delta X) = RS \|\nabla_Y f\|^2 \|\Delta X\|^2 \qquad (42)$$

Following this, the optimal error can be found in a similar manner to the linear layer in (37). Combining (42) with (8), and (7), results in the final solution for a convolution layer:

$$\left( \Delta x_{nchw}^{(*)} \right)^2 = \frac{e^2 V^2}{RSNCHW \|\nabla_Y f\|^2} \qquad (43)$$

### B.3 Advanced Convolution

**Single Layer:** We now add horizontal stride and vertical stride $T$ to the simple convolution. The input and output activation dimensions will become important, so we define the input and output dimensions to be $N \times C \times H_i \times W_i$ and $N \times K \times H_o \times W_o$, respectively. The weight gradient calculation with stride is:

$$\frac{\partial f}{\partial \theta_{kcrs}} = \sum_{n,h_o,w_o}^{N,H_o,W_o} x_{n,c,Th_o+r,Tw_o+s} \frac{\partial f}{\partial y_{nkh_ow_o}} \qquad (44)$$

We calculate the gradient error bound in a similar manner to (41), by using the Cauchy-Schwarz inequality. As well, we use the substitutions $h_i := Th_o + r$ and $w_i := Tw_o + s$ for brevity, which represent the index into the input image height and width. This results in the gradient error bound:

$$\|\Delta \nabla_\theta f\|^2 \leq \sum_{r,s}^{R,S} \left( \sum_{n,c,h_o,w_o}^{N,C,H_o,W_o} \Delta x_{nch_iw_i}^2 \right) \left( \sum_{n,k,h,w}^{N,K,H_o,W_o} \left( \frac{\partial f}{\partial y_{nkh_ow_o}} \right)^2 \right) \qquad (45)$$

To further simplify, we define a *usage number* $u_{hw}$ which satisfies:

$$\sum_{n,c,h,w}^{N,C,H_i,W_i} u_{hw} \Delta x_{nchw}^2 = \sum_{r,s}^{R,S} \sum_{n,c,h,w}^{N,C,H_o,W_o} \Delta x_{n,c,Th+r,Tw+s}^2 \qquad (46)$$

The usage number may seem arbitrary, however, it has the definition as the number of times an activation in the $hw$ position is reused during convolution. Another definition of $u_{hw}$ is the ratio of multiplications for an activation to an equivalent $1 \times 1$ convolution with $T = 1$. We illustrate examples of $u_{hw}$ in Figure S1. The usage can be calculated using the index, activation dimensions, filter dimensions, stride, and padding and under all circumstances $u_{hw} \leq RS$. For all convolutions in this work, we simply use $u_{hw} \approx RS/T^2$. This avoids having multiple compression rates within the same activation channel (requires per-activation bitwidth tracking) and avoids the need to determine compression rates on a per-activation basis (computationally expensive). Convolutions with $T \neq 1$ comprise a small portion of network layers, thus the impact from this approximation is small.

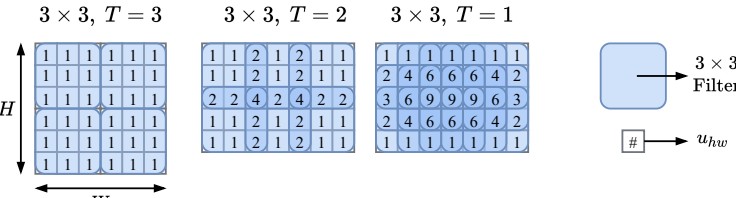

Figure S1: Examples for the usage number $u_{hw}$ with a $3 \times 3$ filter size and various strides $T$. Overlapping areas are used in more than one filter product. Padding is ignored.

Substituting (46) into (45) results in:

$$D(\Delta X) = \|\nabla_Y f\|^2 \sum_{n,c,h,w}^{N,C,H_i,W_i} u_{hw} \Delta x_{nchw}^2 \tag{47}$$

with a constrained optimization solution of:

$$\left(\Delta x_{nchw}^{(*)}\right)^2 = \frac{e^2 V^2}{NCH_iW_iu_{hw}\|\nabla_Y f\|^2} \tag{48}$$

**Multiple Layers:** Residual networks and their derivatives have *skip connections* where an activation is used by more than one layer. We can ignore layers that do not depend on the activation error $\Delta X$, which avoids many cases where this occurs. However, skip connections often involve an activation being used in multiple convolutions; this is why we solve the problem directly where an activation $X$ is used in $L$ convolution layers (Figure S2).

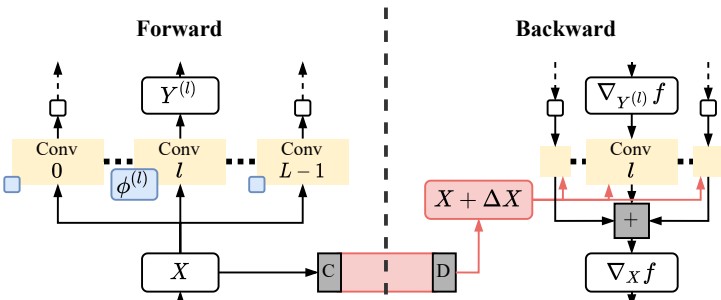

Figure S2: Activation reuse with $L$ convolution layers. Blue: Parameters, Yellow: Layers, Red: Compressed activations, C: Compression, D: Decompression.

In this case, the activation $X$ is used in $L$ convolutions with $L$ weight tensors $\{\theta^{(0)}, ... \theta^{(l)}, ... \theta^{(L-1)}\}$ to produce $L$ activation tensors $\{Y^{(0)}, ... Y^{(l)}, ... Y^{(L-1)}\}$. The issue is that the compressed activation is used by all $L$ convolutions which propagates error to their weights. Hence, we need to consider all layers when determining an error bound for the constrained optimization problem.

If we consider the problem of bounding the gradient error of all layers, $\theta = \{\theta^{(0)}, ... \theta^{(l)}, ... \theta^{(L-1)}\}$, we can split the sums of squares into individual layers:

$$\|\Delta\nabla_\theta f\|^2 = \sum_{l}^{L} \|\Delta\nabla_{\theta^{(l)}} f\|^2 \tag{49}$$

We can create inequalities in a similar manner to the previous problems, particularly the inequality from (45) and the inclusion of a usage number for each layer $l$, $u_{hw}^{(l)}$. Using these

we can find:

$$D(\Delta X) = \sum_l^L \|\nabla_{Y^{(l)}} f\|^2 \sum_{n,c,h,w}^{N,C,H_i,W_i} u_{hw}^{(l)} \Delta x_{nchw}^2 \tag{50}$$

with a constrained optimization solution of:

$$\left(\Delta x_{nchw}^{(*)}\right)^2 = \frac{e^2 V^2}{NCHW \sum_l^L u_{hw}^{(l)} \|\nabla_{Y^{(l)}} f\|^2} \tag{51}$$

The interpretation of (51) is that the error tolerance decreases for each additional layer. This is intuitive, as the activation is being used by more weights, which results in a larger overall gradient error.

### B.4   Batch Normalization

Batch Normalization (BN) presents a host of problems when attempting to bound the error in a compression sensitive context. The equations for BN are quite different from convolution, which makes achieving a closed-form solution for error difficult. BN activation errors propagate to *both* the parameter and activation gradients and then to other layers in the network. This makes bounding the error difficult.

The equations that describe a batch normalization layer are described below. These are transcribed from Ioffe & Szegedy [6]. We define $M := NHW$ for brevity.

| Shape | Forward | Description |
|---|---|---|
| $(N, C, H, W)$ | $x_{nchw}$ : from prev. layer | Input activation |
| $(C,)$ | $\gamma_c$ : parameter | Learned scaling |
| $(C,)$ | $\beta_c$ : parameter | Learned bias |
| $(C,)$ | $\mu_c = M^{-1} \sum_{n,h,w} x_{nchw}$ | Input mean |
| $(C,)$ | $\sigma_c^2 = M^{-1} \sum_{n,h,w} (x_{nchw} - \mu_c)^2$ | Input variance |
| $(C,)$ | $s_c = (\sigma_c^2 + 10^{-5})^{-1/2}$ | Inverse Std. Dev. |
| $(N, C, H, W)$ | $a_{nchw} = s_c(x_{nchw} - \mu_c)$ | Normalized activation |
| $(N, C, H, W)$ | $y_{nchw} = \gamma_c(a_{nchw} + \beta_c)$ | Normalized and scaled output |

| Shape | Backward | |
|---|---|---|
| $(N, C, H, W)$ | $\frac{\partial f}{\partial x_{nchw}} = s_c\gamma_c\left(\frac{\partial f}{\partial y_{nchw}} - M^{-1}\left(a_{nchw}\frac{\partial f}{\partial \gamma_c} + \frac{\partial f}{\partial \beta_c}\right)\right)$ | (52) |
| $(C,)$ | $\frac{\partial f}{\partial \gamma_c} = \sum_{n,h,w}^{N,H,W} \frac{\partial f}{\partial y_{nchw}} a_{nchw}$ | (53) |
| $(C,)$ | $\frac{\partial f}{\partial \beta_c} = \sum_{n,h,w}^{N,H,W} \frac{\partial f}{\partial y_{nchw}}$ | (54) |
| $(N, C, H, W)$ | $\frac{\partial f}{\partial a_{nchw}} = \gamma_c \frac{\partial f}{\partial y_{nchw}}$ | (55) |
| $(N, C, H, W)$ | $\frac{\partial f}{\partial y_{nchw}}$ : from next layer | |

A diagram of activation error propagation in the backward pass is shown in Figure S3 for an example network that consists of CONV 1 $\rightarrow$ CONV 2 $\rightarrow$ BATCH NORMALIZATION. In the backward pass, orders are reversed, and the gradients from the batch normalization layer are passed to the convolution layers. The equations of the BN layer result in activation errors causing errors in the parameter gradient $\nabla_\gamma f$ and the outgoing activation gradient $\nabla_X f$. Simply constraining the parameter gradient $\nabla_\gamma f$ in a similar manner to convolution layers

will not work, as $\nabla_X f$ is usually used in subsequent computations in the backward pass (e.g., CONV 1 and CONV 2, Figure S3). This means that errors from BN layers will propagate to all subsequent layers through the activation gradients (red, Figure S3).

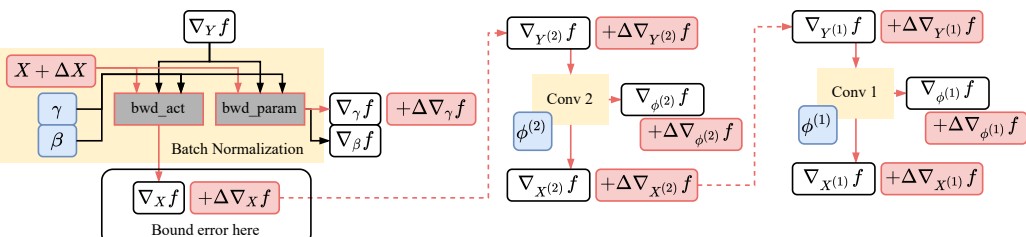

Figure S3: Error propagation in the backward pass resulting from Batch Normalization (BN) activation error. Paths affected from the injected activation error $\Delta X$ are shown in red. As the errors propagate to the activation gradient $\nabla_X f$, they can further propagate to any layer preceeding the batch normalization layer. By bounding the outgoing gradient error $\Delta \nabla_X f$ we aim to also bound any subsequent errors.

One way to address this would be to take into account the errors of all weights and parameters affected by each BN layer, for example bounding all of $\Delta \nabla_\gamma f$, $\Delta \nabla_{\theta^{(2)}} f$ and $\Delta \nabla_{\theta^{(1)}} f$ in Figure S3. Bounding all gradients in this manner requires calculating interactions between layers, which would be computationally expensive. Instead, we treat batch normalization as a sub-problem, i.e., minimizing the loss with respect to $X$ instead of $\gamma$ and $\theta$. By bounding the error of the activation gradient $\Delta \nabla_X f$ we hypothesize that the subsequent errors are also bounded. Unfortunately, the affected computation graph is large and complex, which makes a rigorous proof difficult. Hence, we do not prove convergence bounds for networks with batch normalization.

Consequently, we bound the input activation error as:

$$\|\Delta \nabla_X f\|^2 \leq e^2 V^2 \tag{56}$$

As the equations handling back-propagation in BN layers are complex, a similarly complex derivation is required to arrive at a computationally efficient bound on the activation error. To begin, we find $\frac{\Delta \partial f}{\partial x_{nchw}}$, i.e., the compression error of the derivative w.r.t. $x_{nchw}$ (an element of $\Delta \nabla_X f$). We define $\hat{a}_{nchw} = a_{nchw} + s_c \Delta x_{nchw}$, which can be derived by substituting the activation with error $x_{nchw} + \Delta x_{nchw}$ into the definition of $a_{nchw}$.

$$\frac{\Delta \partial f}{\partial x_{nchw}} = s_c \gamma_c \left( \frac{\partial f}{\partial y_{nchw}} - M^{-1} \left( \hat{a}_{nchw} \left( \frac{\partial f}{\partial \gamma_c} + \frac{\Delta \partial f}{\partial \gamma_c} \right) + \frac{\partial f}{\partial \beta_c} \right) \right)$$
$$- s_c \gamma_c \left( \frac{\partial f}{\partial y_{nchw}} - M^{-1} \left( a_{nchw} \frac{\partial f}{\partial \gamma_c} + \frac{\partial f}{\partial \beta_c} \right) \right) \tag{57}$$

$$= -\frac{s_c \gamma_c}{M} \left( \hat{a}_{nchw} \frac{\Delta \partial f}{\partial \gamma_c} + s_c \Delta x_{nchw} \frac{\partial f}{\partial \gamma_c} \right) \tag{58}$$

with

$$\left( \frac{\Delta \partial f}{\partial x_{nchw}} \right)^2 = \frac{s_c^2 \gamma_c^2}{M^2} \left( \hat{a}_{nchw} \frac{\Delta \partial f}{\partial \gamma_c} + s_c \Delta x_{nchw} \frac{\partial f}{\partial \gamma_c} \right)^2 \tag{59}$$

We use the inequality $(a+b)^2 \leq 2(a^2 + b^2)$ to simplify the square:

$$\left( \frac{\Delta \partial f}{\partial x_{nchw}} \right)^2 \leq \frac{2 s_c^2 \gamma_c^2}{M^2} \left( \hat{a}_{nchw}^2 \left( \frac{\Delta \partial f}{\partial \gamma_c} \right)^2 + s_c^2 \Delta x_{nchw}^2 \left( \frac{\partial f}{\partial \gamma_c} \right)^2 \right) \tag{60}$$

We make one approximation in the solution of batch normalization. Although this approximation is not necessary to obtain a solution, it greatly simplifies the result which helps reduce

the overhead for computation at runtime. We assume $a_{nchw}^2 \ll s_c^2 \Delta x_{nchw}^2$, which allows for $\hat{a}_{nchw}^2 \approx s_c^2 \Delta x_{nchw}^2$. Empirically we find this to be valid, as with moderate compression $\Delta x_{nchw}$ is large, while $a_{nchw}$ is normalized and hence insignificant by comparison. As we are using inequalities, the inequality can still hold true provided that the approximation is close to correct. This allows us to write:

$$\left( \frac{\Delta \partial f}{\partial x_{nchw}} \right)^2 \lessapprox \frac{2s_c^4 \gamma_c^2}{M^2} \Delta x_{nchw}^2 \left( \left( \frac{\Delta \partial f}{\partial \gamma_c} \right)^2 + \left( \frac{\partial f}{\partial \gamma_c} \right)^2 \right) \tag{61}$$

$$\tag{62}$$

The term $\left( \frac{\Delta \partial f}{\partial \gamma_c} \right)^2 + \left( \frac{\partial f}{\partial \gamma_c} \right)^2$ is an expression for the compressed derivative w.r.t $\gamma_c$, and depends on $\Delta X$. As the goal is to keep the the expression quadratic in $\Delta X$, we avoid introducing an third order term by bounding this term with some $g_c^2$ satisfying $(g_c) \leq (\hat{\nabla}_\gamma f)^2$. It is of note that this bounds the compressed parameter gradients in a similar manner to convolution layers, however, the goal is to arrive at a compute-friendly solution for $\Delta \nabla_X f$. In practice, $g_c^2$ can be defined for any method given the maximum compression error.

Finally, we can express the activation gradient error as:

$$\|\Delta \nabla_X f\|^2 \lessapprox \sum_{n,c,h,w}^{N,C,H,W} \frac{2s_c^4 \gamma_c^2}{M^2} g_c^2 \Delta x_{nchw}^2 \tag{63}$$

$$\tag{64}$$

and thus use the gradient error bound

$$D(\Delta X) = \sum_{n,c,h,w}^{N,C,H,W} \frac{2s_c^4 \gamma_c^2}{M^2} g_c^2 \Delta x_{nchw}^2 \tag{65}$$

which when used with the method of Lagrange multipliers (7), results in the activation error bound

$$\left( \Delta x_{nchw}^{(*)} \right)^2 = \frac{2e^2 V^2 NHW}{C \gamma_c^2 s_c^4 g^2} \tag{66}$$

The parameter gradient bound $g_c$ can be derived with knowledge about the maximum activation error, incoming gradient $\nabla_Y f$, standard deviation $s_c$, and value of $\gamma$. We substitute activation error into the equation for the parameter gradient (53) and continue from there:

$$\left( \frac{\Delta \partial f}{\partial \gamma_c} \right)^2 = \left( s_c \sum_{n,h,w}^{N,H,W} \Delta x_{nchw} \frac{\partial f}{\partial y_{nchw}} \right)^2 \tag{67}$$

$$\leq \left( s_c \sum_{n,h,w}^{N,H,W} \max_{nhw} |\Delta X| \max_{nhw} |\nabla_Y f| \right)^2 \tag{68}$$

$$= s_c^2 (NHW)^2 \max_{nhw} (\Delta X)^2 \max_{nhw} (\nabla_Y f)^2 \tag{69}$$

where $\max_{nhw}$ describes a maximum over the batch, height, and width dimensions of the tensor, and $|...|$ denotes the element-wise absolute value.

To tighten the inequality, we make the observation that in nearly all cases of BN the incoming gradient $\nabla_Y f$ is sparse. Many BN layers are followed by a ReLU activation, which introduces sparsity in the gradient. If we define $\kappa_c (\leq NHW)$ as the number of non-zero values in the $c$-th channel of $\nabla Y$, then a tighter bound can be arrived at by reducing the number of sum terms:

$$\left(\frac{\Delta \partial f}{\partial \gamma_c}\right)^2 \leq s_c^2 \kappa_c^2 \max_{nhw}(\Delta X)^2 \ \max_{nhw}(\nabla_Y f)^2 \tag{70}$$

We cannot use $\Delta X$ above to solve (66) to obtain $\Delta X$, thus we need a bound which does not depend on $\Delta X$. If, at maximum compression, any activation becomes zero, then $\max_{nhw}(\Delta X^2) \leq \max_{nhw}(X)^2$. Further, if we are using the signed fixpoint compression, and always remain above 2 bits, it can be derived that:

$$\max_{nhw}(\Delta X)^2 \leq \max_{nhw}(X/2)^2 \tag{71}$$

which holds true for all networks we have examined (generally bitwidths range from 4-10 bits). This results in our choice of $g_c$:

$$g_c := s_c^2 \kappa_c^2 \max_{nhw}(X/2)^2 \ \max_{nhw}(\nabla_Y f)^2 + \left(\frac{\partial f}{\partial \gamma_c}\right)^2$$

## B.5   Layer Normalization

Layer normalization is similar to batch normalization, however statistics are taken over the channel index, rather than the batch index. The equations that describe a layer normalization layer are below. We use a 2D activation tensor, with only batch ($N$) and channel ($C$) indices for this layer. These are transcribed from Ba et al. [3].

| Shape | Forward | Description |
|---|---|---|
| $(N,C)$ | $x_{nc}$ : from prev. layer | Input activation |
| $(C,)$ | $\gamma_c$ : parameter | Learned scaling |
| $(C,)$ | $\beta_c$ : parameter | Learned bias |
| $(C,)$ | $\mu_n = C^{-1}\sum_c x_{nc}$ | Input mean |
| $(C,)$ | $\sigma_n^2 = C^{-1}\sum_c (x_{nc} - \mu_c)^2$ | Input variance |
| $(C,)$ | $s_n = (\sigma_n^2 + 10^{-5})^{-1/2}$ | Inverse Std. Dev. |
| $(N,C)$ | $a_{nc} = s_n(x_{nc} - \mu_n)$ | Normalized activation |
| $(N,C)$ | $y_{nc} = \gamma_c(a_{nc} + \beta_c)$ | Normalized and scaled output |

| Shape | Backward | |
|---|---|---|
| $(N,C)$ | $\frac{\partial f}{\partial a_{nc}} = \gamma_c \frac{\partial f}{\partial y_{nc}}$ | (72) |
| $(N,C)$ | $\frac{\partial f}{\partial y_{nc}}$ : from next layer | |

We leave the remainder of the gradients to derive in this section. The goal is to arrive at an expression for the error on the outgoing activation gradient, $\|\Delta \nabla_X f\|^2$ .The following partial derivatives can be derived from the above equations and the chain rule. $\delta_{ab}$ is the Kronecker delta for $a$ and $b$.

$$\frac{\partial \mu_r}{\partial x_{nc}} = \delta_{nr}\frac{1}{C} \tag{73}$$

$$\frac{\partial y_{rk}}{\partial x_{nc}} = \gamma_k\frac{\partial a_{rk}}{\partial x_{nc}} \tag{74}$$

$$\frac{\partial \sigma_r^2}{\partial x_{nc}} = \delta_{nr}\frac{2}{C}\left(x_{rc} - \mu_r - \frac{1}{C}\sum_k^C x_{rk} - \mu_r\right) \tag{75}$$

$$\frac{\partial s_r}{\partial x_{nc}} = \delta_{nr}\frac{-1}{C}s_r^2\left(a_{rc} - \frac{1}{C}\sum_k^C a_{rk}\right) \tag{76}$$

By definition, the sum of the normalized values must be zero, $\sum_k^C a_{rk} := 0$. We can thus simplify the derivative of the inverse standard deviation:

$$\frac{\partial s_r}{\partial x_{nc}} = \delta_{nr}\frac{-1}{C}s_r^2 a_{rc} \tag{77}$$

By application of the chain rule, we can derive the simplified expression for the gradient with respect to $x_{nc}$.

Beginning with the normalized activation's derivatives:

$$\frac{\partial a_{rk}}{\partial x_{nc}} = \frac{\partial s_r}{\partial x_{nc}}(x_{rk} - \mu_r) + s_r\left(\delta_{rn}\delta_{ck} - \frac{\partial \mu_r}{\partial x_{nc}}\right) \tag{78}$$

$$= \left(\delta_{nr}\frac{-1}{C}s_r^2 a_{rk}\delta_{kc}\right)\left(\frac{a_{rk}}{s_r}\right) + s_r\delta_{rn}\delta_{ck} - s_r\left(\delta_{nr}\frac{1}{C}\right) \tag{79}$$

$$= \frac{-\delta_{nr}\delta_{kc}s_r}{C}a_{rk}^2 + s_r\delta_{nr}\delta_{ck} - \frac{\delta_{nr}s_r}{C} \tag{80}$$

And continuing to the gradient of $x_{nc}$:

$$\frac{\partial f}{\partial x_{nc}} = \sum_{r,k}^{N,C}\frac{\partial f}{\partial y_{rk}}\frac{\partial y_{rk}}{\partial x_{nc}} \tag{81}$$

$$= \sum_{r,k}^{N,C}\gamma_k\delta_{nr}\left(\frac{-s_r}{C}a_{rk}^2\delta_{kc} + s_r\delta_{ck} - \frac{s_r}{C}\right) \tag{82}$$

$$= \gamma_c s_n - \frac{s_n}{C}\gamma_c a_{nk}^2 - \frac{s_n}{C}\sum_k^C \gamma_k \tag{83}$$

Note that as only the activation is compressed, not the values of $\gamma_c$ or $s_n$, their values will not change. Errors will only occur on the compressed version of $a_{nk}$. If the compressed value is $\hat{a}_{nc}$, then the $X$ gradient error is:

$$\left(\frac{\Delta\partial f}{\partial x_{nc}}\right)^2 = \frac{s_n^2\gamma_c^2}{C}\left(\hat{a}_{nc}^2 - a_{nc}^2\right) \tag{84}$$

expanding the squared error:

$$\hat{a}_{nc}^2 - a_{nc}^2 = s_n^2\Delta x_{nc}^2 + 2a_{nc}\Delta x_{nc} + a_{nc}^2 - a_{nc}^2 \tag{85}$$

$$= s_n^2\Delta x_{nc}^2 + 2a_{nc}\Delta x_{nc} \tag{86}$$

and assuming that the activation error is large relative to the normalized activation:

$$\approx s_n^2\Delta x_{nc}^2 \tag{87}$$

results in the final expression for the gradient error:

$$\|\Delta \nabla_X f\| \approx \sum_{n,c} \frac{s_n^4 \gamma_c^2 \Delta x_{nc}^2}{C} \tag{88}$$

This becomes the choice of $D(\Delta X)$ for a layer normalization layer. Proceeding with solution by the method of lagrange multipliers results in an activation error bound of:

$$\left(\Delta x_{nc}^{(*)}\right)^2 = \frac{Ce^2 V^2}{s_n^4 \gamma_c^2} \tag{89}$$

# Appendix C   Algorithm and AutoX Implementations

In this appendix we detail the AC-GC training algorithm, as well as the implementation for the AutoX methods.

## C.1   Training with AC-GC

Algorithm 1 describes the method for training with an AC-GC bounded compression method. In the forward pass, norms are calculated every `recalc_iters` iterations. The quantities vary depending on the layer, and include $\|X\|^2$, $s^2$ and $\gamma^2$. On the **last_use_of** the activation $X$, the last 10 norms calculated are sampled, and either the **mean** or **max** of them is used to produce summary statistics for each quantity. Finally, compression is applied using error bounds calculated using Table 1. In the backward pass, the activation is decompressed, producing an activation with error $\hat{X}$ which is used to compute the lossy gradients $\hat{\nabla}_X f$ and $\hat{\nabla}_\theta f$. These are then used in the next layer, and to update the network parameters (not shown).

## C.2   AutoQuantZ Implementation

For AutoQuant we use scaled fixpoint compression from [5], Fixpoint activation compression has been presented in a variety of ways [4, 7]. We use per-channel scaling which appears to have the best accuracy [5]. The conversion for signed values is:

$$f_c := \max_{nhw}|X| \tag{90}$$

$$q_{nchw} := \text{int}(2^{b-1}x_{nchw}/f_c, b) \tag{91}$$

where $q_{nchw}$ is the fixpoint activation with $b$ bits, $f_c$ is the maximum of $|x_{nchw}|$ for the channel $c$, and $\text{int}(..., b)$ converts the value to a signed integer of $b$ bits. This prevents clipping of activations, as the scaled value never exceeds the fixpoint range.

The bitwidth for such signed integer compression can be calculated using

$$b \geq -\log_2 \Delta x_{nchw} - \log_2 f_c$$

For an unsigned activation, we can remove one bit (the sign bit) and obtain the same error.

Zero Value Compression is a relatively simple compression mechanism where sparse values are replaced with a bitmask representing non-zero values, and non-zero values compressed separately [11]. During activation decompression, the entire activation is decompressed at once, allowing for the entire activation to be compressed without indices to determine the locations of the compressed indices. We re-implement ZVC in our code.

## C.3   AutoCuSZ Implementation

We adapt the CuSZ implementation from [13]. As the implementation uses an error bound directly during compression, there is little modification required to input the error bounds from Table 1. We re-implement this work in AutoCuSZ, and opt for 1D compression to avoid compressing across multiple GPU cache lines. The compression is configured with a radius of 4096, and 8-bit encoding, which we empirically find gives highest compression rates.

## C.4   AutoJPEGZ Implementation

To create AutoJPEGZ we begin with the implementation of JPEG activation compression from [5]. This implementation uses two Discrete Quantization Tables (DQTs) for early and late training conditions. The DQT describes the amount of loss in the encoded frequency coefficients. JPEG-ACT tuned the 64 DQT coefficients using CIFAR10/ResNet50 and found that flatter DQTs provide better accuracies and activation compression ratios for activations.

JPEG does not have a bounded error for a given DQT, thus we attempt to approximately bound the error using an empirical accuracy-compression relationship. We first parameterize a flat $8 \times 8$ DQT as follows:

**Algorithm 1** Pseudocode for training a DNN using AC-GC and a generic compression method. Differences between BatchNormalization and Convolution are omitted for clarity.

**Inputs:**
    Input activation: $X$
    Compression method with **compress**() and **decompress**() functions
    Sequence of network layers:
    Recalculation interval: recalc_iters
    summary method: **summarize** $\in \{\textbf{mean}, \textbf{max}\}$

1: # Lossless Forward Pass
2: **for** layer **in** layers **do**
3:    $Y = $ layer.**fwd**$(X)$                         ▷ Normal forward pass

4:    **if** t % recalc_iters = 0 **then**           ▷ Early stats saving
5:       **append_norms_for_layer**(layer, $X$)

6:    **if** **last_use_of**$(X)$ **then**          ▷ Compression and cleanup
7:       sample = **get_last_10_norms**(layer)
8:       stats = **summarize**(sample)
9:       $\Delta X = $ **error_bound**(layer, stats)
10:      layer.compressed_X = **compress**$(X, \Delta X)$
11:      **delete** $X$

12:    $X = Y$

13: # ... Loss and initial gradient $\nabla_Y f$ calculated
14: # ... Final activation $Y$ retrieved

15: # Lossy Backward Pass
16: **for** layer **in** **reversed**(layers) **do**

17:    **if** **first_use_of**$(\hat{X})$ **then**              ▷ Decompression
18:       $\hat{X} = $ **decompress**(layer.compressed_X)
19:      **delete** layer.compressed_X

20:    $\hat{\nabla}_X f = $ layer.**bwd_act**$(\hat{X}, Y, \nabla_Y f)$      ▷ Backward pass
21:    $\hat{\nabla}_\theta f = $ layer.**bwd_param**$(\hat{X}, Y, \nabla_Y f)$

22:    **if** t % recalc_iters == 0 **then**        ▷ Late stats saving
23:      **append_norms_for_layer**(layer, $\hat{\nabla}_X f$, $\nabla_Y f$, $\hat{\nabla}_\theta f$)

24:    **if** last_use_of$(Y)$ **then**               ▷ Cleanup
25:      **delete** $Y, \nabla_Y f$

26:    $Y, \nabla_Y f = \hat{X}, \hat{\nabla}_X f$

$$DQT = \begin{bmatrix} d/4 & d \ ... & d \\ d & d \ ... & d \\ & ... & \\ d & d \ ... & d \end{bmatrix} \tag{92}$$

where $d$ is an integer, and the first element is decreased to reduce error on the mean of the activations.

Using this parameterization, we investigate the $d$/error relationship, where increasing $d$ increases compression (Figure S4). We measure this by first sampling uncompressed training of CIFAR10/ResNet50. Every 10 epochs, we save one batch (N=128) worth of activations (30 samples total). Offline, we run JPEG compression on the sampled activations for each value of $d$ to establish the compression-error relationship (Figure S4). While there is a wide range of compression errors (from $2^{-16}$ to $2^0$), we can bound most activations by using a percentile of this distribution. Then, we train all models, using this empirical compression-error relationship to calculate the JPEG compression levels, which approximately satisfy the AC-GC convergence constraint.

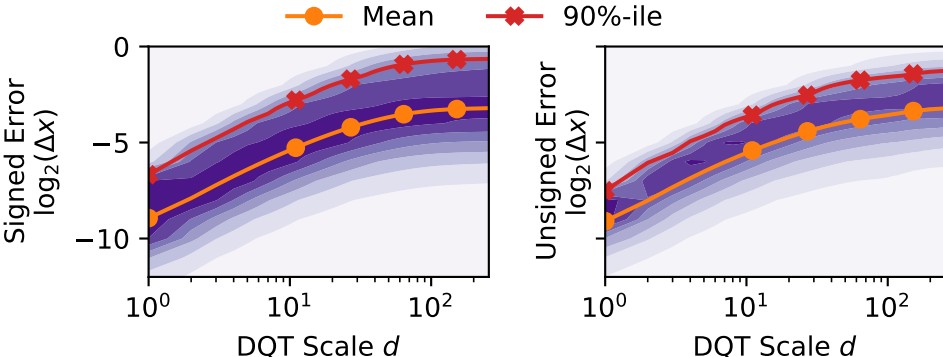

Figure S4: JPEG error response with a scaled constant DQT on activations sampled from every 10 epochs of training CIFAR10/ResNet50. Allowable values for $d$ are integers in the range $[1, 255]$.

We select the 90%-ile for our work as a proof of concept, as our goal is not to rigorously study JPEG compression, simply to present a case study for how AC-GC error bounds can be used with unbounded compression methods.

# Appendix D   Detailed Hyperparameters

In this section, we summarize the dataset and model setup for this work.

We use the CIFAR10 [2] and IMDB.fine [10] dataset implementations [14], and use the standard training/test dataset split. We use ImageNet 2012[9], with the validation dataset, as we do not have access to the test dataset. ImageNet images are scaled so that the shortest edge is 256 pixels. From the Div2K [1] datasets, we perform $2\times$ upsampling from bicubic downsampled images (LR_BICUBIC_2X $\rightarrow$ HR). Div2K images are cropped to the minimum dimensions present in the dataset.

Dataset augmentations are shown in Table S1, and selections of hyperparameters are shown in Table S2. Models have no significant modifications from their original source.

Table S1: Dataset augmentations for this work.

| Dataset | Augmentations |
|---|---|
| **CIFAR10** | RANDOM SCALE UP TO $1.2\times$, RANDOM CROP TO $32 \times 32$, RANDOM HORIZONTAL FLIP, 10% RANDOM PCA, $15^o$ RANDOM ANGLE, $8 \times 8$ RANDOM CUTOUT |
| **Div2K** | RANDOM CROP TO $48 \times 48$, RANDOM HORIZONTAL FLIP |
| **IMDB.fine** | NO DATASET AUGMENTATIONS |
| **ImageNet** | RANDOM CROP TO $224 \times 224$, RANDOM HORIZONTAL FLIP, 10% RANDOM PCA |

Table S2: Training hyperparameters. $N$ batch size, $lr$ learning rate, $E$ epochs. If not otherwise stated, all use MomentumSGD with a momentum of 0.9. H. Param. Source specifies how hyperparameters were chosen.

| Model | $lr$ | $N$ | $E$ | $lr$ DECAY/ $E$ INTERVAL | MODEL / H. PARAM. SOURCE |
|---|---|---|---|---|---|
| **All CIFAR10** | 0.05 | 128 | 300 | $0.5\times$/70 | FROM [12] |
| **Div2K/VDSR** | 0.1 | 32 | 110 | $0.1\times$/50 | RE-IMPLEMENTED / GRID SEARCH |
| **IMDB/CNN** | 0.001 | 64 | 20 | (ADAM) | FROM [14] EXAMPLES |
| **ImageNet/ResNets** | 0.1 | 64 | 105 | $0.1\times$/25 | FROM [14] EXAMPLES/ [15] |

# Appendix E   Extended Experimental Results

To accompany the experimental section in the main text, we include tabulated data for all experiments. Table S3 shows the accuracy resulting from grid search over QuantZ bitwidths. The selected bitwidth (within 0.1 of the baseline) is bolded. We do not show high-bitwidth configurations as they have a similar accuracy to 8-bit fixpoint. QuantZ compression rates are shown in Table S4, which are averages over all runs and all epochs of training. Finally, Table S5 shows accuracy for the AutoX methods with standard deviations included.

Table S3: Full accuracy with standard deviations for all models with fixpoint compression (QuantZ). Bolded accuracy was selected for GridQuantZ as accuracy is within 0.1 of the baseline. Configurations of 10, 12, 14, and 16 bits are not shown as accuracy is similar to 8-bit for all models. All non-ImageNet configurations are run five times.

| Model | Base | 2-bit QuantZ | 4-bit QuantZ | 6-bit QuantZ | 8-bit QuantZ |
|---|---|---|---|---|---|
| **CIFAR10** % Top-1 Test Accuracy | | | | | |
| VGG | 93.6±0.11 | 89.2±0.73 | 93.4±0.16 | 93.4±0.11 | **93.5±0.09** |
| RN50 | 94.9±0.12 | 85.6±2.47 | 94.5±0.07 | **95.0±0.09** | 95.1±0.12 |
| WRN | 95.8±0.04 | 90.9±1.42 | 95.6±0.10 | **96.0±0.06** | 95.8±0.04 |
| **Div2K** Best Val. PSNR | | | | | |
| VDSR | 36.1±0.00 | 31.7±0.37 | 35.8±0.03 | **36.0±0.00** | 36.1±0.01 |
| **IMDB** % Best Val. Accuracy | | | | | |
| CNN | 61.4±0.20 | **61.7±0.35** | 61.6±0.30 | 61.3±0.20 | 61.5±0.37 |
| LSTM | 60.3±0.52 | **60.4±0.45** | 60.4±1.27 | 60.2±0.85 | 60.0±0.52 |
| **Text Copy** % Best Test Accuracy | | | | | |
| TRANS | 98.8±0.41 | 9.3±7.87 | 98.3±0.13 | **98.9±0.48** | 99.1±0.10 |
| **ImageNet** % Top-1 Single Crop Val. Accuracy | | | | | |
| RN18 | 68.6±0.17 | 46.6±30.1 | 67.3±0.19 | 68.1±0.67 | **68.5±0.39** |
| RN50 | 72.6±0.28 | 17.4±6.1 | 70.8±0.11 | 72.2±0.22 | **72.5±0.38** |

Table S4: Average compression rates for all models with fixpoint compression (QuantZ). Selected compression rates for each network are bolded.

| Model | 2-bit QuantZ | 4-bit QuantZ | 6-bit QuantZ | 8-bit QuantZ |
|---|---|---|---|---|
| **CIFAR10** | | | | |
| VGG | 15.9× | 10.5× | 7.9× | **6.3×** |
| RN50 | 12.5× | 7.8× | **5.7×** | 4.4× |
| WRN | 15.5× | 10.2× | **7.6×** | 6.1× |
| **Div2K** | | | | |
| VDSR | 14.2× | 9.1× | **6.7×** | 5.3× |
| **IMDB** | | | | |
| CNN | **16.5×** | 11.1× | 8.3× | 6.7× |
| LSTM | **14.7×** | 7.6× | 5.2× | 3.9× |
| **Text Copy** | | | | |
| TRANS | 14.0× | 7.5× | **5.1×** | 3.9× |
| **ImageNet** | | | | |
| RN18 | 9.2× | 5.4× | 3.8× | **2.9×** |
| RN50 | 13.4× | 8.5× | 6.2× | **4.9×** |

Table S5: Full accuracy with standard deviations for all models with AutoX compression. N/A indicates that the configuration was not examined. All ImageNet configurations are run three times, and the remaining configurations are run five times.

| Model | AutoQuantZ | AutoCuSZ | AutoJPEGZ |
|---|---|---|---|
| **CIFAR10** % Top-1 Test Accuracy | | | |
| VGG | 93.5±0.11 | 93.5±0.09 | 92.9±0.17 |
| RN50 | 95.0±0.11 | 94.7±0.19 | 94.3±0.57 |
| WRN | 95.9±0.20 | 95.8±0.06 | 95.3±0.11 |
| **Div2K** Best Val. PSNR | | | |
| VDSR | 36.1±0.01 | 35.8±0.08 | 36.1±0.00 |
| **IMDB** % Best Val. Accuracy | | | |
| CNN | 61.6±0.33 | 61.8±0.38 | 61.4±0.58 |
| RNN | 60.1±0.81 | 60.9±0.92 | N/A |
| **Text Copy** % Best Test Accuracy | | | |
| TRANS | 98.6±0.54 | 98.3±0.10 | N/A |
| **ImageNet** % Top-1 Single Crop Val. Accuracy | | | |
| RN18 | 68.5±0.24 | 68.1±0.14 | 68.1±0.29 |
| RN50 | 72.7±0.18 | 72.5±0.57 | 71.5±1.47 |

## Appendix F    Additional Experiments

### F.1    Sensitivity Study

In this appendix, we examine the empirical relationship between $e$ and compression, as well as whether using separate parameterizations for different layer types is beneficial. As bounds for convolution and batch normalization layers are obtained using differing procedures, it is useful to examine if there is a differing sensitivity with each layer type. We train CIFAR10/ResNet50 using AutoQuant and the training parameters listed in the primary manuscript. For convenience, we define

$$\epsilon := -\log_2 |e| \tag{93}$$

We sweep across a grid $\epsilon$ values for the convolution and BN layers and measure average accuracy, loss, and bitwidth (across five randomly initialized runs).

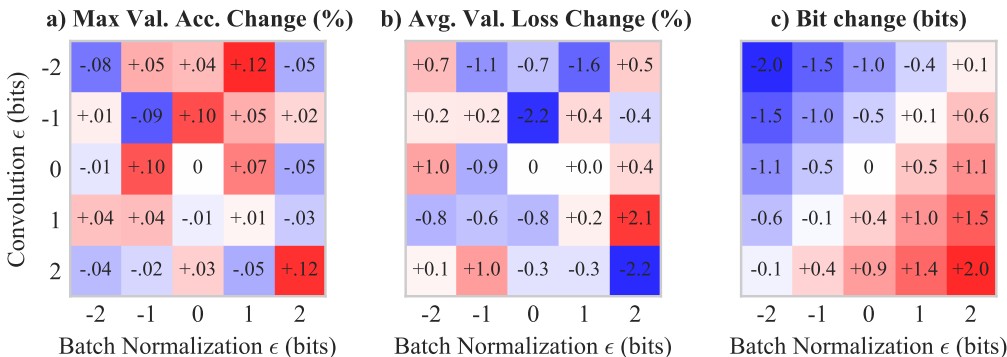

Figure S5: Sensitivity to $\epsilon$ for CIFAR10/ResNet50. a) Maximum validation accuracy over the course of training, b) Average training loss over the final 10 epochs, c) Average bitwidth over the course of training. All values are the average of 5 training runs, and are represented relative to convolution $\epsilon = 0$ and batch normalization $\epsilon = 0$.

The sweeps demonstrate the correlation where increasing $\epsilon$ results in increased accuracy (Figure S5a), but decreased bitwidth (Figure S5c). The average bitwidth over training

changes nearly identically with $\epsilon$, except for slight deviations due to an uneven split of activations between convolution and BN. Examining accuracy across constant-bitwidth diagonals reveals that BN layers have a slightly lower error tolerance than convolution layers (Figure S5a top-right is higher than bottom-left). Despite the noise in the results, it also appears that the -0.5 bits diagonal has higher accuracy than the 0 bits diagonal (+0.09% on average), likely due to non-convexity of the error response. These results demonstrate that the AutoQuant bitwidth is within 0.5 bits and 0.12% of a locally-optimal solution.

### F.2 Norm Approximation Error

In this section, we examine the error distribution resulting from approximating the norms for the AC-GC error bounds. To collect this data, we train ImageNet/ResNet50 using AutoQuantZ. Every 10 epochs for 100 iteration, the exact and approximated (mean and max) norms are measured, and the error between them is recorded. We measure across all layers in the network.

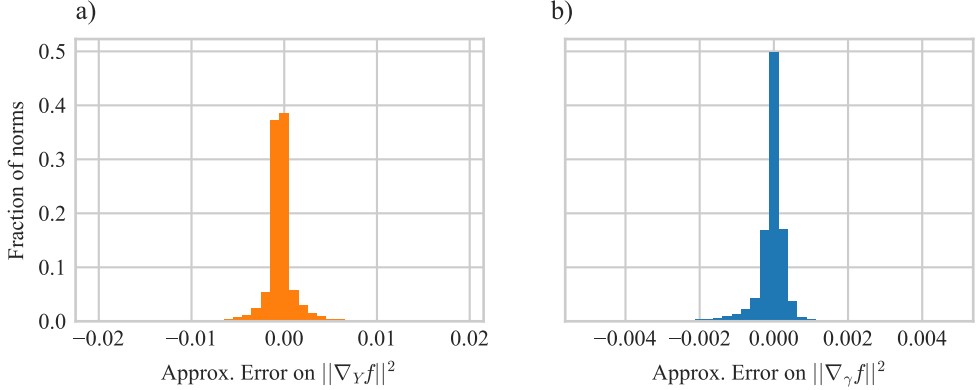

Figure S6: Histogram of mean approximation error for the activation gradient $\|\nabla_Y f\|^2$ and batch norm gradient $\|\nabla_\gamma f\|^2$ for the first 100 epochs of ImageNet/ResNet50.

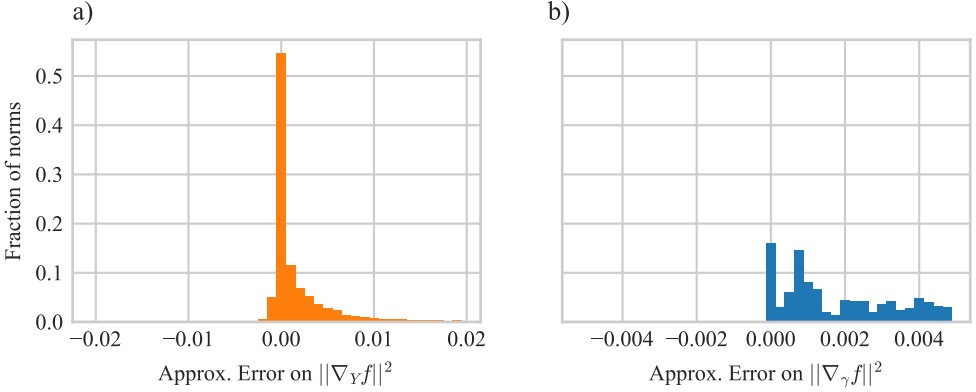

Figure S7: Histogram of max approximation error for the activation gradient $\|\nabla_Y f\|^2$ and batch norm gradient$\|\nabla_\gamma f\|^2$ for the first 100 epochs of ImageNet/ResNet50.

In Figure S6 it can be seen that the approximations have residuals with approximately zero mean, and a normal distribution with small variance. This is advantageous, as it indicates that there is little difference from measuring norms at every iteration. Overall, we find that the mean is an over-estimate of the norms, as the norms follow a distribution with a long tail.

In Figure S7 it can be seen that using a maximum instead of mean results in biased residuals. However, as all norms are *over-approximated*, compression rate will decrease somewhat.