# OpenReview forum: "AC-GC: Lossy Activation Compression with Guaranteed Convergence"
_NeurIPS.cc/2021/Conference — NeurIPS 2021 Poster_

### Official Review · Reviewer_H5sM · 2021-07-12

**Rating:** 7
**Confidence:** 3

**Summary:**

The authors bound the increase in loss in the case that activations used for backpropagation have errors. By doing so, they can set a fixed acceptable level of increase in the loss, and solve for the error level that can be tolerated for each activation used in backpropagation (and therefore deduce a level of compression). This analysis is relatively agnostic to the precise implementation of the compression. They evaluate their approach on a variety of datasets, including relatively large datasets such as ImageNet, demonstrating that their method can achieve high compression levels with relatively little tuning.

**Limitations And Societal Impact:**

I think it is fine as it is.

**Main Review:**

### Detailed Feedback

Abstract: I think the authors did a good job here. Motivated well, and clear description.

Intro: The motivation is good, and well explained (as with the abstract). I think that fig1 is also nicely presented.

Preliminaries:
- Throughout the paper, you seem to use $\phi$ for weights, which is a little unusual. Might be better to use $\theta$ to be more consistent with other works?
- A purely stylistic observation: It might be good to move the related work here, and motivate why what you're doing is different to that. This is my preference (but I am not going to consider this with regard to scoring)
- L91: There's a comment here about compression not slowing down optimization too much; I note that this claim is made later too. But what evidence is there for this assertion?

S3:
- This entire section is written fairly well, with nice figures and useful intuition provided. But the one thing I would note is that I am _not_ an expert on this area, and as such I find it difficult to provide much technical criticism -- it all seems reasonable to me. Nothing stands out to me as being obviously questionable.

S4:
- This section was more difficult to follow than the last one -- it is very dense. If it's possible to provide any figures it would be very appreciated by readers.
- It might be a good idea to discuss information theory and its relation to compression, along with lagrange multipliers in the preliminaries?
- L199: the discussion of max pooling has been forgotten (although it is very simple).

S5:
- L210: could expand on how you collected this data?

S6
- The caption for fig5 is not useful, and you have to search for the interpretation in the text. You may want to push the explanation to the caption.
- The results in table 2 are great. It would be appreciated if you could add some results for training time though :-)
- The results for table 3 are less impressive, unfortunately (although they have been achieved in far less time -- you undersell that in the table). Also, what about AutoCuSZ and AutoJPEG? The results don't seem particularly impressive when compared with [1]
- (I don't expect this for rebuttal) It'd be really great if the authors could look at providing some experiments using a transformer. This might be a fair bit of work -- such as deriving bounds for layer norm -- but would make this work more complete.
- L287: can you provide numbers for how long compression takes?

S7:
- It would be a good idea to cite ActNN [1] that was recently published at ICML 21.

### Overall Thoughts

I quite like this paper overall, as there is a (seemingly) sensible theoretical analysis, and it is generic to the particular choice of compression. The problem area is extremely important, and this work will be of wide interest. I think the only real downside is that the results obtained are not favorable when compared to [1] (at least on ImageNet), although it is not possible to make a direct comparison. The authors may be able to deal with this by swapping in a different compression backend on the ImageNet experiments. I would ask the authors to address some of the concerns above -- especially the ones regarding overheads. I am sympathetic that the implementation is likely to be far from optimal, but it is important information to provide.

==== Post rebuttal

I'm happy with the author's response, and I have increased my score accordingly :-)

[1] ActNN: Reducing Training Memory Footprint via 2-Bit Activation Compressed Training

**Time Spent Reviewing:**

3

---

> ### Author Response · Authors · 2021-08-10
> **Response to Reviewer H5sM**
>
>
> We found your detailed comments particularly helpful for identifying ways to improve the quality of our manuscript. Below, we will address your main questions and concerns.
>
> ### Preliminaries
>  1. **On weight notation:** We agree that $\theta$ is a more standard choice and will update our manuscript to use it.
>
>
>  3. **On compression overheads:** Compression/decompression can generally be performed in parallel with the CNN forward/backward pass. Provided there are resources available for both workloads, there will theoretically be no slowdown. See GIST [2], where performance overheads were 4% (parallel compression/compute), vs. approx. 30% for ActNN [1] (sequential compression/compute) for similar reduced precision compression.
>
> ### S4
>  1. **On improving clarity:** We can add another figure that details the differences between the constraint, maximum error, and how bitwidth results from these quantities. The changes we describe for Reviewer coLe (see **Clarity** questions 1. and 2.) should also help address this.
>
>
>  2. **On information theory and Lagrange Multipliers:** We present an example using Lagrange Multipliers in Appendix B.1, and will add a discussion on information theory to the descriptions on lossy compression. The rate-distortion tradeoff is particularly relevant to our work.
>
>
> 3. **On max pooling:** Max pooling requires a bitmask to store the locations of the maximum values. We’ve added this to the main text.
>
>
> ### S5
>  1. **On collection of L210 data:** We have a description of JPEG error bounding in Appendix C.4. The sampling procedure involves uncompressed training of CIFAR10/ResNet50. Every 10 epochs, we save one batch (N=128) worth of activations (30 samples total). Offline, we run JPEG compression to establish the compression-error relationship (Supplemental Figure S4). Then, we train all models, using this empirical compression-error relationship to calculate the JPEG compression levels, which approximately satisfy the error bounds in Table 1.
>
>
> ### S6
>  1. **On Fig. 5 interpretation:** We agree that this is unclear, and will move the description into the Fig. 5 caption.
>
>
>  2. **On training time:** Our current implementation is un-optimized, and performance is largely determined by compression/decompression speed. Our primary contribution, AC-GC error bound calculation, is 0.4% of total training time. For ImageNet/ResNet50 and batch size 128, AutoQuantZ performance is 1.66x vs. naive swapping to the CPU,  and 0.64x vs. uncompressed training with batch size 32.
>
>
>  3. **On Table 3 results:** The poor results in Table 3 resulted from using single runs of ImageNet in the original submission (i.e., some lucky/unlucky seeds). We have now performed 3 duplicate training runs, showing that on ResNet50, AutoQuantZ (+0.2% acc., 4.8x compr.) is superior to GridQuantZ (-0.1% acc., 4.8x compr.). ResNet18 results are similar to ResNet50. We will add standard deviations for ImageNet (0.2% to 0.3%) to the supplemental material. This highlights the difficulty in performing grid search, as duplicate runs are required to obtain high quality results.
>
> Additionally, we have examined AutoJPEGZ, and AutoCuSZ on ImageNet. ResNet50 has acc. and compr. of -0.2% and 10.1x (AutoCuSZ), and -1.1% and 8.5x (AutoJPEGZ). AutoCuSZ is a significant improvement over grid search (GridQuantZ).
>
> * **On ActNN:** ActNN[1] and AC-GC approach activation compression from opposite perspectives. For ActNN, they find the best loss/accuracy for a given compression rate with fixpoint compression. For AC-GC, we find the best compression rate for a given loss/accuracy change for any error bounded compression method. Both perspectives have important use cases. The ActNN approach works well when the goal is to fit within a memory budget. Our approach allows for selecting the target loss increase *a priori*, and gaining information on the uncompressed accuracy from a single training run. On an unknown model, AC-GC will also converge similarly to the uncompressed case, providing assurances that loss changes are due to model/ hyperparameter configuration, not compression. Additionally, ActNN is specific to group-wise fixpoint compression, whereas AC-GC is quick to adapt to any lossy compression method. We obtain similar compression rates on most models, for example, on ImageNet/ResNet50, we achieve acc. and compr. of -0.1% and 10.1x for AutoCuSZ, vs. -0.2% and 12x for ActNN-L3.
>
>
>  4. **On Transformers:** We have Transformer and LSTM results in the response to Reviewer coLe (see **Models**). We obtain low accuracy changes, and better compression than grid search. The bounds for layer norm are similar to batch normalization:
>
> \begin{equation}
>     \Delta x^{(\ast)2}_{nchw} = \frac{C e^2 V^2}{2 s_n^4 \gamma_c^2}
> \end{equation}
>
> Where $s_n$ is the per-batch inverse standard deviation, and the remaining quantities are equivalent to their batch normalization counterparts.
>
>
>  5. **On how long compression takes:** See S6-2 (On training time)
>
> ### S7
>  1. **On ActNN citation:** We will include the discussion from S6-3 (On ActNN) in the next draft, as well as quantitative performance, compression, and accuracy comparisons.
>
> ### References
> [1] J. Chen et al., ActNN: Reducing Training Memory Footprint via 2-Bit Activation Compressed Training, ICML 2021
>
> [2] A. Jain et al., Gist: Efficient Data Encoding for Deep Neural Network Training, ISCA 2018

---

> > ### Comment · Reviewer_H5sM · 2021-08-26
> > **Score Raised**
> >
> > Nice job on the rebuttal. I raised my score a little while ago, but just to confirm to other reviewers + the AC -- I moved up to a 7.

---

> > > ### Author Response · Authors · 2021-08-27
> > > **Thank you**
> > >
> > > Thank you! We are happy to have addressed your comments and will include the suggested changes in the next version of the paper.

---

### Official Review · Reviewer_7589 · 2021-07-14

**Rating:** 7
**Confidence:** 5

**Summary:**

The paper proposes a theoretical convergence upper bound for DNN training with compressed activation. Using this bound, the compression rate is controlled over training epochs by computing the quantization error. Moreover, to reduce the complexity of computing the activation’s quantization error, the paper uses statistics of this error such as mean and max instead of computing the complex norm. The efficiency of this new approach (AC-GC) is evaluated in various benchmarks including the ImageNet dataset and compared with previous approaches such as WAGE, GIST, and JEPG-ACT. The result showed that the AC-GC compressed the activation more than other approaches while maintaining accuracy compared to FP32.


**Ethical Concerns:**

There are not ethical issues with this paper

**Limitations And Societal Impact:**

The authors adequately addressed the limitations and potential negative societal impact of their work

**Main Review:**

Strength:

1- The idea behind the constraint (activation precision) convergence rate under the Polyak-Łojasiewicz Condition is novel and interesting. The author uses a common Lagrange multiplier approach to optimize the constraint.

2- The paper is well organized. The paper technically sounds correct and claims well supported by theoretical analysis and experimental results.

3- The proposed approach is evaluated in large and variant benchmarks, which mean the proposed approach can be deployed for different models

Weakness:

1-  The author did not compare the new approach with previous papers such as [1] and [2], where the theatrical convergence rate is proposed for low-precision DNN parameters and activations. It would be suggested the comparison with these works is discussed in the paper.

2-  The author needs to add more clarity on the motivation behind storing the activation ( not cited two frameworks ) and provide the assumptions to satisfy the Polyak-Łojasiewicz Condition in the paper.
3- Is this the convergence rate depends on the dimension of activation? How does the dimension of activation affect the compression rate?

4- The last sentence of the abstract is a bold statement, and I would suggest changing it. The AC-GC has shown valuable results for the experiments presented in the paper, and it is hard to conclude that the approach works for any model.

5-  It would be suggested to compute and report the approximation error using the max and mean of activation error instead of an accurate norm.

References:

[1] Li, Z., & De Sa, C. (2018). Dimension-free bounds for low-precision training (https://papers.nips.cc/paper/2019/hash/d4cd91e80f36f8f3103617ded9128560-Abstract.html)

[2] Li, Hao, et al. "Training quantized nets: A deeper understanding." Proceedings of the 31st International Conference on Neural Information Processing Systems. 2017.

Post Rebuttal ======
Thanks for the clarification. I think the response resolved my comments on the clarity and quality part. I raised my score from 6 to 7. I encourage authors to add their responses to my comments to the various parts of the papers to improve the quality of the paper. I also suggest that emphasize the weak assumption of convexity in the abstract or introduction.



**Time Spent Reviewing:**

24

---

> ### Author Response · Authors · 2021-08-10
> **Response to Reviewer 7589**
>
> We are glad you found AC-GC novel and interesting. Below we address the weaknesses mentioned.
>
> ### Weaknesses
>  1. **On comparison with [1] and [2]:** Training Quantized Nets [2] presents an interesting theoretical framework for fully quantized training. We found their examination of exploration vs. exploitation especially intriguing. Similarly, Dimension-free bounds [1] improves this framework by removing the dependence on weight dimension by bounding the quantization variance. The convergence bounds for [1] & [2] require strong convexity, whereas the PL condition used in our work is a weaker assumption. While both AC-GC and these works start with convergence bounds, they are used very differently. Our work focuses on compressed activation training, where lossy activations are used *only in the backward pass*, whereas these works use quantized values in *both the forward and backward pass*. The advantage of compressed activation training in our work is that errors are only accumulated in the backward pass, improving accuracy. We agree both these works are relevant and will add them to our discussion of related work. We compare favorably against the state-of-the-art in 4-bit fully quantized training in Table 2 (ULP, or Ultra-Low Precision Training).
>
>
>  2. **On motivation to store activations:** The primary reason for storing activations is to avoid recomputing in the backward pass. As the backward pass proceeds in reverse, a large portion of the network would need to be recomputed, significantly increasing training times. You are correct that some works [3,4] have proposed ways of avoiding activation storage, however, they require specific network structures, limiting utility.
>
> * **On assumptions for PL condition:** The assumptions for convergence by Karimi et al. [5] are currently stated in Appendix A. Briefly, the assumptions are that the loss is continuous, differentiable, has bounded variance, and $\mathbb{E}[\nabla_\phi f] = \nabla \mathcal{L}$, and the PL Condition. These are similar to [1,2], with the use of the PL Condition instead of strong convexity. The PL Condition is a weaker assumption than strong convexity, and is, for some $\eta$ and all $\phi$:
>
> \begin{equation}
>     \frac{1}{2} \lVert \nabla_\phi \mathcal{L}(\phi) \lVert^2
>         \geq \eta (\mathcal{L}(\phi) - \mathcal{L}(\phi^{(*)}))
> \end{equation}
>
>
>  3. **On convergence and dimension:** You are correct that convergence rate (as defined by Karimi et. al [5]) is dependent on activation dimension. As batch size ($N$) increases, gradient variance decreases $V^2$ proportional to $1/N$, improving convergence. The effect of batch size on compression is an interesting avenue, however we have not examined it. Ignoring the effect of convergence rate, fixpoint bitwidth $b$ should scale as $b \propto \log_2(N)$.
>
>
>  4. **On working with any model:** We agree and will change the final sentence in the abstract. A more precise statement is that our method “is suitable to any model composed of the layers analyzed, with an average accuracy change of 0.10% for the models studied”. As well, convergence under these conditions and assumptions is theoretically guaranteed. However, in practice there are other factors which can prevent convergence, for example floating point errors. We will add a discussion in the conclusions on other factors which can affect convergence and accuracy.
>
>
>  5. **On approximating norms:** We will add statistics to the supplemental material on the approximation error from using the max and mean to approximate norms. The change in bitwidth from using either the max, mean, or exact norm is small, approx. 0.5-1.0 bits for any activation. As well, most differences average out over a few iterations.
>
> ### References
>
> [1] Li, Z., & De Sa, C.  Dimension-free bounds for low-precision training, NeurIPS 2019
>
> [2] Li, Hao, et al. Training quantized nets: A deeper understanding, NeurIPS 2017
>
> [3] A. Gomez et al. The Reversible Residual Network: Backpropagation Without Storing Activations, NeurIPS 2017
>
> [4] T. Chen et al. Training Deep Nets with Sublinear Memory Cost, ArXiv [1604.06174] 2017
>
> [5] H. Karimi et al.  Linear Convergence of Gradient and Proximal-Gradient Methods Under the Polyak-Łojasiewicz Condition, ML and KD in DBs 2016

---

> > ### Comment · Reviewer_7589 · 2021-08-26
> > **Post Rebuttal**
> >
> > Thanks for the clarification. I think the response resolved my comments on the clarity and quality part. I raised my score from 6 to 7. I encourage authors to add their responses to my comments to the various parts of the papers to improve the quality of the paper. I also suggest that emphasize the weak assumption of convexity in the abstract or introduction.

---

> > > ### Author Response · Authors · 2021-08-27
> > > **Thank you**
> > >
> > > We are glad we resolved your comments. We will emphasize the assumptions about loss conditioning and convexity in the introduction.

---

### Official Review · Reviewer_coLe · 2021-07-16

**Rating:** 6
**Confidence:** 4

**Summary:**

This paper studies activation compression to reduce the memory consumption when training models with stochastic gradient descent. The goal is to establish an activation compression framework with convergence guarantees, while eliminate the tuning overhead from conventional search based method for compression precision selection. The contributions of this paper come in three folds:

1. The authors first established a connection between the activation compression error and the convergence bound (i.e. the upper bound on training loss).

2. Built on the above theory, the authors formulate the activation compression precision selection problem as a constrained optimization problem, which can accommodate typical DNN layers such as linear layers, convolution layers and normalization layers.

3. Empirically, the authors combine this optimization-based activation precision selection method with various compression methods (with bounded errors at each compression precision). They show that it could attain matching accuracy (minimal acc degradation compared to no activation compression) compared to conventional grid search method for activation precision without the grid search (i.e. hyperparameter tuning) overhead.

**Ethical Concerns:**

I do not see anything improper or having ethical issues with this paper.

**Limitations And Societal Impact:**

I think the attached checklist answers the questions adequately. The overhead / limitation of the method is discussed in Section 6.

**Main Review:**

Originality:

Previous activation compression works typically use time-consuming hyperparameter tuning methods to select compression precision. This paper studies the connection between activation error and convergence, and explores a novel approach to automatically select activation compression precision. I think the theoretical analysis, the methods brings new thoughts and methodology to the area.



Quality:

My major concern/question on the paper is about the claim that the proposed activation precision selection approach *does not need any hyperparameter tuning* to achieve minimal accuracy loss.  The basis of claiming no tuning is to uniformly use a fixed e^2 value (e^2 = 1 is used in the paper) across models and datasets. Theoretically, the limit of the loss bound in equation (13) can increase by 50% if e^2 = 1 is introduced due to activation compression, which could trigger significant loss / accuracy degradation; such a case can be instantiated using a quadratic function.  Totally understand e^2 = 1 could potentially be a good hands down option for a broad range of SOTA models. But to validate this, it needs more model/datasets (such as transformer based models for vision and NLP) using e^2=1 for activation precision while the current experiments only cover convolutional models.

Besides the major above question, it would be great if the authors can resolve the following questions in the rebuttal and paper draft:

1. Theorem 1 (line 141) assumes the expectation of the gradient error norm is zero. Does this means the theorem assumes there is no gradient error at all (because the assumption is only satisfied when the activation compression does not trigger any gradient change)? If this is the case, the assumption is too unrealistic. Also from the proof in the appendix A, it is unclear how this assumption is leveraged in the proof.

2. For the constrained problem in equation (7), the presentation needs to be improved to clearly define the problem. From the proofs in Appendix B, I believe equation (7) is actually referring to \Delta X* = \min_{b} B(\Delta X), s.t. D(\Delta X) < e^2V^2 / 2. The current formulation is not rigorously reflecting the optimization problem.

3.  (Not a major concern) In table 3, it seems the grid searched QuantZ is giving 0.7% higher accuracy on ImageNet than the no compression baseline. Is it because of random seed variation? Or is it because the no compression baseline is not well tuned to optimize validation accuracy (The typical ResNet 50 ImageNet validation top 1 accuracy is around or around 75% to 80% [1] while the baseline results in the paper is giving 72-73% top 1 accuracy)?


Clarity: I think the paper explain intuitions and general ideas very well. But technical clarity (such as terminology definition and etc.) should be improved in several places.

1. In equation 3, is \Delta x_{nchw} a function of compression precision b. If this is the case, it is unclear how this function (defined on discretized supports) is used in the continuous constrained problem in equation (7).

2. In line 161, the range of \Delta X is not properly defined if \Delta X is a multi dimensional vector. My guess is you mean a ball centered at 0 with radius \Delta X*.

3. In line 252, there is a claim on "all networks examined have loss changes below 2%". It needs a bit clarification in the text because the loss change at -log|e| is definitely larger than 2% given loss changes from ~0.015 to ~0.025.


Significance: If the proposed methods can show strong accuracy in broader models / tasks (such as NLP or vision transformers) still with a uniform value for hyperparameter e^2, the impact can be significant to eliminate / dramatically reduce the tuning overhead on compression precision selection.


I am willing to raise the score if the above concerns are resolved.


Reference

[1] Deep Residual Learning for Image Recognition. He et. al.


**Time Spent Reviewing:**

5 hours

---

> ### Author Response · Authors · 2021-08-10
> **Response to Reviewer coLe**
>
> Thank you for your valuable feedback on AC-GC. We will address all comments in the next version of the paper.
>
> ### On claims of no hyperparameter tuning
> The following clarifies our claims of “no hyperparameter tuning”. This claim does *not* stem from the use of $e^2=1$ across most of our experiments. Instead, our claim results from the guarantee that the relationship between the compressed and uncompressed loss is known *a priori*. Imagine performing hyperparameter tuning using compressed training with an unknown DNN. Suppose that some hyperparameter configurations result in a significant loss increase. With existing approaches, the poor loss could be due to hyperparameter configuration or compression, with no way to determine which. AC-GC gives an *a priori* bound on the degree to which the compression can affect the loss, such that the effect of compression can be removed from hyperparameter tuning. While our bound is relatively loose, obtaining this information with existing methods would require multiple runs at different compression rates. In practice, our goal is to select the desired loss increase once, and focus on tuning other hyperparameters, decreasing overall training time.
>
> However, we do agree that the above does not precisely correspond to "no hyperparameter tuning". We will modify the final manuscript with the above motivating example, and clarify that our bound provides information that is only attainable by running existing methods multiple times.
>
>
> ### Models
> We now have additional results for an LSTM on IMDB (AutoQuantZ -0.1% acc., 10.0x compr.). We also have results for a Transformer on a copy dataset, i.e., translation from a sequence to the same sequence shifted by one token (similar to [3][4]). We use this toy dataset due to the short rebuttal time frame. AC-GC works similarly well on the Transformer (AutoQuantZ -0.01% acc., 7.5x compr.), demonstrating the applicability of our method to multiple models (with $e^2=1$). We achieve similar or better compression at the same accuracies as grid search (GridQuantZ 9.8x compr. on LSTM,  3.4x compr. on Transformer).
>
> ### Questions
>
> 1. **On assumption of error norms:** Our apologies, the assumption for Theorem 1 (line 141) is a typo. The correct assumption is $\mathbb{E}[\Delta \nabla_\phi f] = 0$, i.e., the gradient error is unbiased. We do not use this assumption as it is only required for convergence according to Karimi et al.[1]. This is equivalent to assuming $\mathbb{E}[\hat{\nabla}_\phi f] = \mathbb{E}[\nabla_\phi f] = \nabla \mathcal{L}$. We can provide a proof that this assumption holds for linear and convolution layers if requested.
>
>
>  2. **On constrained optimization formulation:** Our optimization finds the error $\Delta X^* \in \mathbb{R}^{N \times C \times H \times W}$ which corresponds to the maximum compression (approximated by $B(\Delta X)$) and satisfies the convergence constraint. Note that we are maximizing over the error $\Delta X$, not bitwidth $b$. A more rigorous definition is:
>
> \begin{equation}
>     \Delta X^* = \underset{\Delta X}{\textrm{argmax}} \ B(\Delta X) \quad \textrm{s.t.} \quad D(\Delta X) \leq e^2 V^2 / 2
> \end{equation}
> Using the Method of Lagrange Multipliers on this problem results in the original Eqn. (7). We will replace Eqn. (7) in the manuscript with this optimization problem definition.
>
> 3. **On above-baseline accuracies:** The above-baseline accuracies in Table 3 were due to using single runs of ImageNet in the original submission (i.e., some lucky/unlucky seeds). We have now performed 3 duplicate training runs for ImageNet, with a ResNet50 acc. and compr. of -0.1%  and 4.8x for QuantZ, and +0.2% and 4.8x for AutoQuantZ.  This highlights the utility of AC-GC, as this kind of variation can make locating a suitable compression-accuracy tradeoff difficult. However, we observe higher-than-baseline accuracies for IMDB (Table 2), where we hypothesize a regularization effect is occurring due to increased gradient noise [2].
>
> * **On low ImageNet accuracy:** Our ImageNet accuracies are lower as we do not use random scaling (which improves performance), and we report 1-crop accuracy. Our 10-crop accuracy for ResNet50 is 75.2%.
>
> ### Clarity
>
> 1. **On discrete optimization** and 2. **On $\Delta X$ definition:** We address these questions below by clarifying our optimization. The AC-GC error bound is found using:
>
> \begin{equation}
>     \Delta X^* = \underset{\Delta X}{\textrm{argmax}} \ B(\Delta X) \quad \textrm{s.t.} \quad D(\Delta X) \leq e^2 V^2 / 2
> \end{equation}
>
> Where the maximum error $\Delta X^* \in \mathbb{R}^{N \times C \times H \times W}$ maximizes compression and satisfies the convergence constraints.  In response to 1, the optimization is over the error $\Delta X$, which is a continuous domain. We then calculate the precision (or other compression parameters) using the error bound.
>
> In response to 2, L161 attempts to describe that the error of the method $\Delta X$ must be less than $\Delta X^*$. This is better described as follows: any $\Delta X \in \mathbb{R}^{N \times C \times H \times W}$ satisfies the convergence constraint provided that all $(\Delta X)^2 \leq (\Delta X^*)^2$. This is because the $D(\Delta X)$’s we select are convex with a minimum at $\Delta X = 0$.
>
> For fixpoint, as we can only select integral bitwidths, we select the highest bitwidth $b$ such that the maximum error is less than $\Delta X^*$. As the actual errors will be smaller than $\Delta X^*$, we lose some potential compression. Calculating $b$ from $\Delta X^*$ is straightforward, however, we do not go into much detail as it depends on the scaling and fixpoint method. Our specific formulation is in Appendix C.2 and in the provided code.
>
> We will modify the text to include descriptions similar to the above.
>
>  3. **On claims of network loss:** This should read "all networks examined **with $e^2=1$ have** loss changes below 2%"
>
> We are triple-checking the main text and supplemental material to ensure that all descriptions are clear and concise. Thank you for your comments!
>
> ### References
>
> [1] H. Karimi et al.,  Linear Convergence of Gradient and Proximal-Gradient Methods Under the Polyak-Łojasiewicz Condition, ML and Knowledge Discovery in Databases 2016
>
> [2] A. Neelakantan et al., Adding Gradient Noise Improves Learning for Very Deep Networks, ArXiv [1511.06807] 2015
>
> [3] A. Graves et al., Neural Turing Machines, ArXiv [1410.5401] 2014
>
> [4] D. Britz et al., Efficient Attention using a Fixed-Size Memory Representation, ArXiv [1707.00110] 2017

---

> > ### Comment · Reviewer_coLe · 2021-08-22
> > **Thanks for the clarification**
> >
> > Thanks for the clarification and additional results.
> >
> > I think the response resolved my comments on the clarity and quality part.
> >
> > I raised my score from 5 to 6.

---

> > > ### Author Response · Authors · 2021-08-27
> > > **Thank you**
> > >
> > > Thank you and we are pleased that our response was satisfactory. We will incorporate your feedback into the next version of the paper.

---

### Decision · Program_Chairs · 2021-09-27

**Decision:**

Accept (Poster)

**Comment:**

While there was a discrepancy between the overall scores initially, the authors' rebuttal letter clarified several concerns of the reviewers, which resulted in all the scores being positive. Hence, I recommend an acceptance. Please implement all the changes suggested by the reviewers in the camera-ready version.